# In situ ultrastructures of two evolutionarily distant apicomplexan rhoptry secretion systems

Shrawan Kumar Mageswaran [1], Amandine Guérin[2,4], Liam M. Theveny[1,4], William David Chen [1,4], Matthew Martinez[1], Maryse Lebrun[3], Boris Striepen[2] & Yi-Wei Chang [1✉]

Parasites of the phylum Apicomplexa cause important diseases including malaria, cryptosporidiosis and toxoplasmosis. These intracellular pathogens inject the contents of an essential organelle, the rhoptry, into host cells to facilitate invasion and infection. However, the structure and mechanism of this eukaryotic secretion system remain elusive. Here, using cryo-electron tomography and subtomogram averaging, we report the conserved architecture of the rhoptry secretion system in the invasive stages of two evolutionarily distant apicomplexans, *Cryptosporidium parvum* and *Toxoplasma gondii*. In both species, we identify helical filaments, which appear to shape and compartmentalize the rhoptries, and an apical vesicle (AV), which facilitates docking of the rhoptry tip at the parasite's apical region with the help of an elaborate ultrastructure named the rhoptry secretory apparatus (RSA); the RSA anchors the AV at the parasite plasma membrane. Depletion of *T. gondii* Nd9, a protein required for rhoptry secretion, disrupts the RSA ultrastructure and AV-anchoring. Moreover, *T. gondii* contains a line of AV-like vesicles, which interact with a pair of microtubules and accumulate towards the AV, leading to a working model for AV-reloading and discharging of multiple rhoptries. Together, our analyses provide an ultrastructural framework to understand how these important parasites deliver effectors into host cells.

[1] Department of Biochemistry and Biophysics, Perelman School of Medicine, University of Pennsylvania, Philadelphia, PA, USA. [2] Department of Pathobiology, School of Veterinary Medicine, University of Pennsylvania, Philadelphia, PA, USA. [3] LPHI, UMR 5235 CNRS, Université de Montpellier, Montpellier, France. [4] These authors contributed equally: Amandine Guérin, Liam M. Theveny, William David Chen. ✉email: yi-wei.chang@pennmedicine.upenn.edu

ntracellular parasites of the phylum Apicomplexa such as *Plasmodium*, *Cryptosporidium* and *Toxoplasma* cause important diseases around the globe including malaria, severe diarrhea, and encephalitis with a particularly strong impact on young children[1–3]. Motile stages of these pathogens share a molecular assembly, the apical complex, which harbors two types of specialized secretory organelles called micronemes and rhoptries[4–8]. First, the contents of micronemes are secreted onto the parasite surface to mediate host cell attachment and gliding motility, which in turn stimulates secretion of rhoptry contents into the host cell to facilitate invasion and establishment of infection[5,6,9–13]. The number of rhoptries varies according to species—*Cryptosporidium spp.* have one, *Plasmodium spp.* have two and *Toxoplasma spp.* have eight-to-twelve. Rhoptries have a unique club-shape morphology and resident proteins are spatially segregated into two distinct structural regions—an anterior tubular neck and a posterior bulb positioned deeper within the cell body[14–17]. Proteins of the rhoptry neck are predominantly involved in invasion while proteins of the bulb are required to establish the vacuole membrane that surrounds the parasite in the host cytoplasm[6,18,19], rewire host transcription and act in host immune evasion[20–24]. It is unclear how the peculiar shape of the rhoptry is maintained, and whether this shape contributes to either protein segregation or secretion. A widely accepted model suggests that rhoptry content proteins are directly secreted into the host cytoplasm during parasite invasion[20,25]. The proteins therefore have to be relocated across three membranes—the rhoptry membrane, the parasite plasma membrane, and the host plasma membrane. The mechanism by which this is accomplished is also largely unknown despite the recent expansion of the molecular repertoire for this process—several non-discharge (Nd) proteins crucial for rhoptry secretion have been identified in *T. gondii* and *P. falciparum* by homology search using ciliate counterparts[26].

Previous ultrastructural studies in *T. gondii* and other apicomplexans have reported the presence of an apical vesicle (AV)[26–30] and an apical rosette (with eight peripheral intramembranous particles surrounding a central element)[26,29,31]. Preliminary inspections of *T. gondii* by cryo-electron tomography (cryo-ET) with a modest sample size (20 cells) have revealed interactions between the AV and the rhoptry tips, as well as interactions between the AV and the apical rosette, which together mediate rhoptry tip docking at the parasite plasma membrane and may play a role in secretion[26]. However, several important questions remain unanswered—(i) How is the defined shape of rhoptry neck maintained? (ii) Are the apparent roles of the AV and the apical rosette in mediating rhoptry tip docking conserved among apicomplexans, or even in *T. gondii* tachyzoites? (iii) If the apical rosette does indeed mediate the secretion of rhoptry contents across the membranes of the AV and the parasite plasma membrane, how does its structural organization support this function? and (iv) How are multiple discharge events achieve in apicomplexans like *T. gondii*, which possess several rhoptries?

Here we performed large-scale, high-resolution cryo-ET imaging of two evolutionarily distant apicomplexans (namely *C. parvum* and *T. gondii*). We report highly organized rhoptry contents including helical filaments in the rhoptry neck that closely interact with the membrane and may impart their cylindrical shape. Furthermore, we report the consistent presence of an AV and an apical rosette in both organisms; the latter is a part of a larger proteinaceous feature, which we named the rhoptry secretory apparatus (RSA). Subtomogram averaging of the RSA in both organisms revealed elaborate ultrastructures consisting of similarly organized extracellular, transmembrane and intracellular components; these components could concertedly operate to regulate apicomplexan rhoptry discharge in an evolutionarily conserved fashion, as they appear to mediate fusion of the AV and the plasma membrane. Depletion of Nd9, a protein required for rhoptry secretion[26] caused disruption of the RSA ultrastructure as well as anchoring of the AV in *T. gondii*, thereby beginning to reveal the mechanistic roles for molecular players in rhoptry secretion. Moreover, in *T. gondii*, we noted the presence of multiple AV-like vesicles lined on a pair of intraconoidal microtubules (seemingly directed towards the AV), suggesting a working model for a reloading mechanism to enable successive secretion of multiple rhoptries.

## Results and discussion

**Cryo-electron tomography resolves the apical end of apicomplexans.** Cryo-electron tomography (cryo-ET) visualizes cellular structures in a near-native, frozen-hydrated state and reveals both membranes and proteins in three dimensions but is severely limited by the thickness and the inherent low contrast of biological samples. Previous cryo-ET imaging on *Plasmodium* was therefore limited although it provided important information pertaining to motility and host invasion[32,33]. To circumvent these issues, we performed cryo-ET on the tapered apical ends of relatively thin *C. parvum* sporozoites (Fig. 1a) and *T. gondii* tachyzoites (Fig. 1e) using contrast enhancing imaging technologies (see Methods; we note that the contrast in our tomograms was sufficient to render any computational denoising unnecessary). In both species we were able to resolve and reconstruct parasite pellicle and apical complex along with its different cytoskeletal and membranous components in great detail (see Supplementary Movies 1 and 2). While *C. parvum* showed a single rhoptry, 2-6 rhoptries were observed in *T. gondii* proximal to the apical end. A pair of rhoptries in *T. gondii* and the single rhoptry in *C. parvum* approached the apical tip from the cell body through the apical complex[17,34] comprising of an apical polar ring, the conoid (a specialized tubulin-based barrel-like structure), and at least two conoidal rings (Fig. 1b, f and Supplementary Figs. 1b and 2a; orientation of 2-dimensional (2-D) sections through 3-dimensional (3-D) tomograms is illustrated in Supplementary Fig. 1a). Similarly, micronemes approached the apical tip in both organisms in an ordered fashion. Micronemes of *C. parvum* showed distinct filaments lining the luminal side of their bounding membranes, a feature not observed in *T. gondii* (Supplementary Fig. 3).

**Luminal helical filaments shape rhoptry necks.** We found filaments lining the luminal surface of the rhoptry neck membrane in both *C. parvum* and *T. gondii* ($n > 25$ cells for each organism; Fig. 1b, f and Supplementary Figs. 1c–f and 2b, e show representative images; tomogram sections without overlaid annotations are included in the Source Data file). Careful examination of raw tomograms in both organisms revealed their helical geometry. In *C. parvum*, we noted two conformations, a two-start (Fig. 1c) and a one-start helix (Supplementary Fig. 1d), both right-handed; while the latter is constituted by a single linear filament, the former is composed of two such filaments—shown in orange and blue—that alternately contribute to the helical turns. The helical turns were closely spaced suggesting lateral interactions between subsequent turns. The rhoptry neck was often bent (without discontinuity in the filament; Fig. 1b) and occasionally branched (introducing filament rearrangements at the branch points; Supplementary Fig. 1f), all of which point to plasticity of the filament. In the filament-lined regions, the rhoptry diameter was uniform (mean ± std = $83 \pm 11$ nm and 95% confidence interval or C.I. = 82–83 nm; Fig. 1b and Supplementary Fig. 1g, h), supporting the idea that the helical filaments adopt a preferred

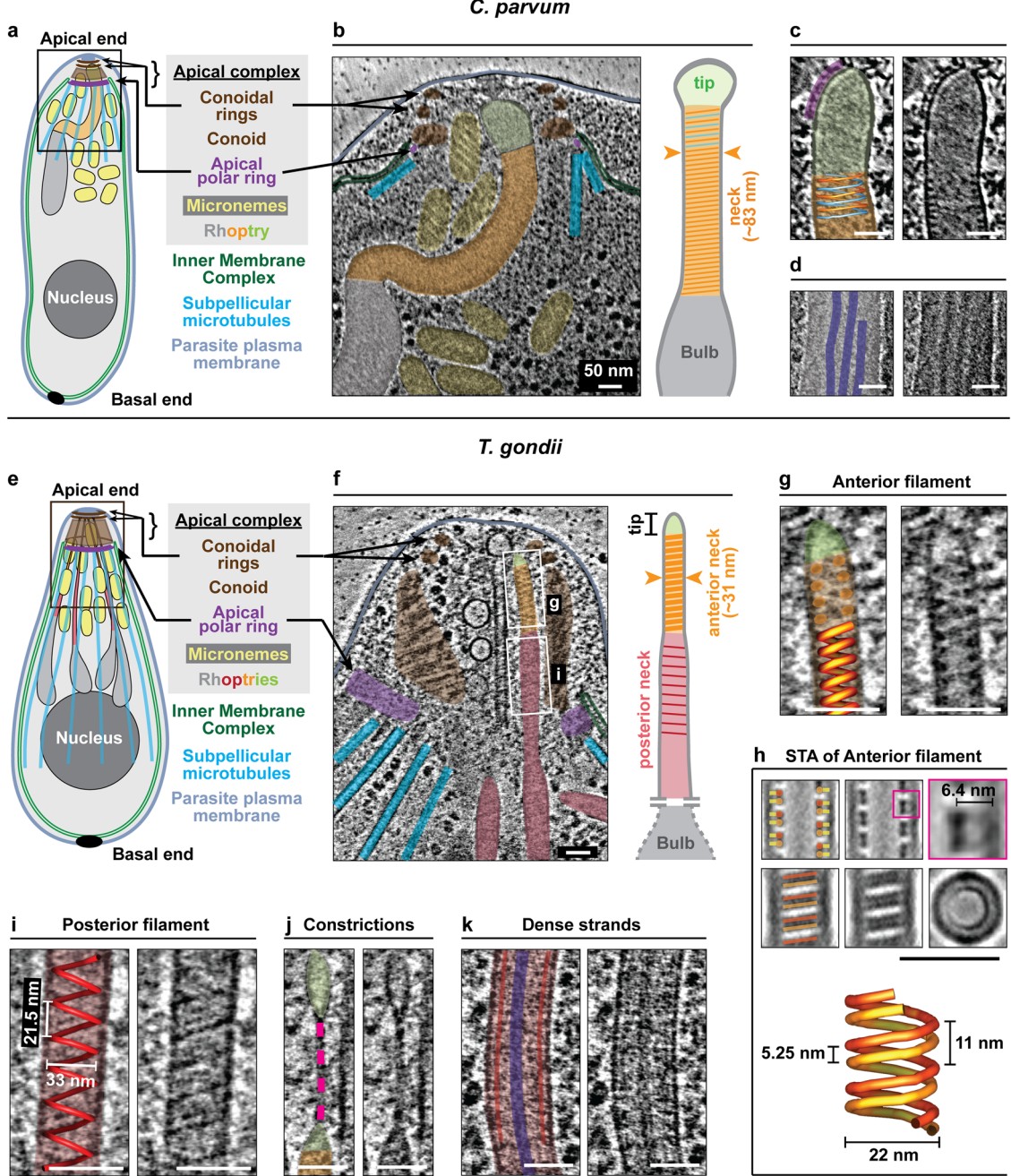

curvature thereby shaping the rhoptry neck. This uniformity in rhoptry diameter was even more evident from individual rhoptries (Supplementary Table 1). The bulb, by comparison, lacked an intact filamentous lining (Fig. 1d and Supplementary Fig. 1e). In *T. gondii*, anterior and posterior neck filaments could be differentiated from raw tomograms (Fig. 1f); the anterior neck filament had a highly regular one-start right-handed helix (pitch of ~11 nm; Fig. 1g, Supplementary Fig. 4, and Supplementary Movie 3) and lined the rhoptry membrane of a consistent diameter, which prompted us to perform subtomogram averaging to increase signal over noise and thereby reveal finer details[35]. Doing so (using two independent sets of tomograms contributing >900 subtomograms each; see Methods and Supplementary Fig. 5) resolved two tightly associated strands of the filament each forming a helix of 22 nm width and 11 nm pitch (Fig. 1h) with clear connections to the membrane. The posterior neck filament, on the other hand, showed a left-handed single helix, which was

marginally wider and more flexible in width and pitch (Fig. 1i and Supplementary Fig. 2b). The opposite handedness of the anterior and posterior neck filaments makes them unlikely to be different conformations of the same filament or composed of the same proteins. Additionally, *T. gondii* rhoptry necks often showed significant local constrictions devoid of luminal filaments (Fig. 1j and Supplementary Fig. 2c), supporting the role of filaments in shaping rhoptry necks and is suggestive of the dynamic nature of these filaments. Similar to *C. parvum*, in *T. gondii*, filamented regions of the rhoptry showed consistent widths (~31 nm for anterior filamented region and ~40 nm for the posterior filamented region; Supplementary Fig. 2e–g). Taken together, membrane-associated rhoptry filaments of different helical arrangements provide a structural principle that may control organelle shape in both species. In addition to neck filaments, the rhoptries of both organisms showed additional structural features including a distinct apical tip (green; Fig. 1b, c, f, g and

**Fig. 1 Luminal filaments shape the rhoptry neck in *C. parvum* and *T. gondii*. a–d** Organization of the single rhoptry in *C. parvum*. **a** Schematic of the parasite including the apical complex. The boxed region is imaged by cryo-ET and shown in **b**. **b** A tomogram slice of the parasite apical end showing organization of the rhoptry and the surroundings. The rhoptry shows three distinct regions – a short anterior tapering tip (green), a middle slender and elongated neck (orange) and posterior broader bulb (gray). Presence of filaments in the rhoptry neck correlates with a uniform diameter while the tip and bulb are devoid of filaments. **c** The rhoptry filament depicted is a two-start right-handed helix (shown with orange and blue individual filaments that alternately contribute to helical turns) while a one-start right-handed helix was also observed (Supplementary Fig. 1d). Other organizational features include **c** a discontinuous layer of protein (purple) at the tip surface, and **d** dense luminal strands (dark blue) predominantly found in the bulb. **e–k** Organization of rhoptries in *T. gondii*. **e** Schematic of the parasite. **f** A tomogram slice of the apical end showing organization of the rhoptries—tip, anterior neck (orange) and posterior neck (red); the bulb is deeper within the cell and outside of the field of view. Anterior and posterior necks show differently organized helical filaments and a correspondingly different rhoptry width. **g** Enlarged image from **f** showing the anterior neck lined with a right-handed helical filament. **h** Subtomogram average (STA) of the anterior neck and filament. Top row: the central slice of the STA from the side view (left and middle) and an enlarged view (right) of the two tightly associated strands (red and orange) of the filament both interacting with the membrane (yellow); middle row: a slice close to the rhoptry surface from the side view showing the two tightly associated strands of the filament as parallel lines (left and middle) and a slice of the STA from the top view (right); bottom row: a 3-D model of the anterior filament. Details of averaging scheme are provided in the Methods section and outlined in Supplementary Fig. 5. A second independent average for the same filament and a combined average are included in Supplementary Fig. 5h, i. 3-D volumes for all averages are included in the repository referenced under "Data availability". **i** Enlarged image from **f** showing posterior neck and associated left-handed helical filament with a 3-D depiction (red helix). **j** A constriction (dashed pink line) in front of the rhoptry neck. **k** A dense luminal strand (dark blue line) in the posterior neck. **j** and **k** are not in the field of view of **f**. Although a helical rhoptry neck filament is not clearly resolved in **k**, a proteinaceous lining (red) close to the rhoptry membrane is suggestive of one, and the uniform width of this rhoptry region suggests that it is the posterior neck. Source data for this figure are provided as a Source Data file. Scale bars in all panels are 50 nm.

Supplementary Fig. 1c) and dense luminal strands (dark blue; Fig. 1d, k and Supplementary Figs. 1c–e and 2d, e); the latter may be constituted by phase separated material or otherwise organized contents and in the case of *T. gondii*, were segregated into the rhoptry neck region of ~60 nm diameter behind the posterior filamented region (zone 3; Supplementary Fig. 2e). Thus our tomograms reveal an overall stepwise organization of structural features along the length of the rhoptry and confirmed the absence of other physical barriers in the rhoptry lumen, especially at the transition from neck to bulb (Supplementary Fig. 1e). If and how these segregated luminal features are related to sub-compartmentalization of rhoptry proteins and their possible role in staged rhoptry secretion and function[19] remains to be established.

**An apical vesicle facilitates rhoptry docking.** We observed an apical vesicle (AV) participating in rhoptry docking at the plasma membrane in all the tomograms of both *T. gondii* and *C. parvum* ($n > 100$ and >150, respectively; Fig. 2a, f). Strikingly, in *C. parvum*, the single rhoptry was fused with the posterior end of the AV such that their bounding membranes were continuous (Fig. 2c, Supplementary Fig. 6a). The luminal passage between the compartments was continuous in some cases, but blocked (marked by *; Supplementary Fig. 6a) in others, suggesting possible regulation of the flow of rhoptry contents into the AV. A "collar" of densities was visible around this membrane connection (Fig. 2c and Supplementary Fig. 6a), which was more evident after subtomogram averaging (Supplementary Fig. 6b). Unlike *C. parvum*, *T. gondii* cells imaged had two rhoptries simultaneously docked but not fused with the AV (21 out of 25 cells used for quantifications, Fig. 2f, h and Supplementary Fig. 6h, i). The rhoptries were associated with the AV at a distance of $10.7 \pm 5.1$ nm (95% C.I. = 9.2–12.2 nm) via densities at their tips (Supplementary Fig. 6j), likely representing a poised pre-fusion state. In both *C. parvum* and *T. gondii*, the AV displayed a characteristic shape and size (teardrop shaped in *C. parvum* while close to ellipsoid in *T. gondii*) and was rigidly anchored onto the apical plasma membrane (Fig. 2b, e, g, j and Supplementary Fig. 6d–g, k–n) at the position of an apical rosette (see details below). In *T. gondii*, we occasionally found an AV with no docked rhoptry (2 out of 25 cells) or only a single docked rhoptry (2 out of 25 cells), with one or two rhoptries nearby (Supplementary Fig. 6i). These may represent assembly or disassembly

intermediates and suggest that the AV anchoring to the plasma membrane is likely independent of rhoptry docking to the AV. Altogether, the AV is a consistent component of the rhoptry system in these evolutionarily distant apicomplexans, and able to coordinate docking of a single rhoptry in *C. parvum* and at most a pair of rhoptries in *T. gondii*. The presence of an AV likely necessitates multiple membrane fusion events for rhoptry secretion, i.e., one between the rhoptry and the AV (as already demonstrated in *C. parvum*) and another between the AV and the parasite plasma membrane.

**Microtubule-associated vesicles may support successive rhoptry secretion.** *T. gondii* tachyzoites have multiple rhoptries (estimated at 8–12[6]) that might allow for successive secretion during multiple invasion attempts[6] or content injection into multiple host cells to alter immune responses without invasion[36]. Relevantly, our tomograms showed the AV accommodating up to two rhoptries simultaneously while several proximal 'free' rhoptries appeared ready and waiting to be docked onto an AV for successive rounds of secretion; these free rhoptries showed similar features to the docked organelles—including tip densities, constrictions and filaments of similar span/length (Supplementary Fig. 7a, b). Such a model requires either reusing or regenerating the AV following secretion of the docked pair. Interestingly, *T. gondii* possesses vesicles that form a line in the intraconoidal space in close proximity to the docked rhoptries[27,28]. In our *T. gondii* tomograms, we confirmed the presence of 3-5 such vesicles (Fig. 2f and Supplementary Fig. 7c, d) and observed their close association with the pair of intraconoidal microtubules (IMTs[37]; similar in their appearance to subpellicular microtubules (Supplementary Fig. 8a, b)); the vesicles line up on only one of the two IMTs via linker-like densities (Supplementary Fig. 7e, f). We therefore refer to them as microtubule-associated vesicles (MVs). The pair of IMTs within each cell showed matching lengths while their lengths varied from cell to cell (Supplementary Fig. 8c). We noted fibrous densities associated with IMTs (Supplementary Fig. 8d) that resembled protofilaments observed during microtubule growth/shortening[38]. We found MVs to have similar shape and size to AVs (Supplementary Figs. 6k and 7g, i), and to be coated with a similar discontinuous layer of protein densities (Supplementary Fig. 7j, k). Furthermore, we observed cases of MVs associating with free rhoptries via the latter's tip densities (2 out of 52 cells; Supplementary Fig. 7l), illustrating their ability for

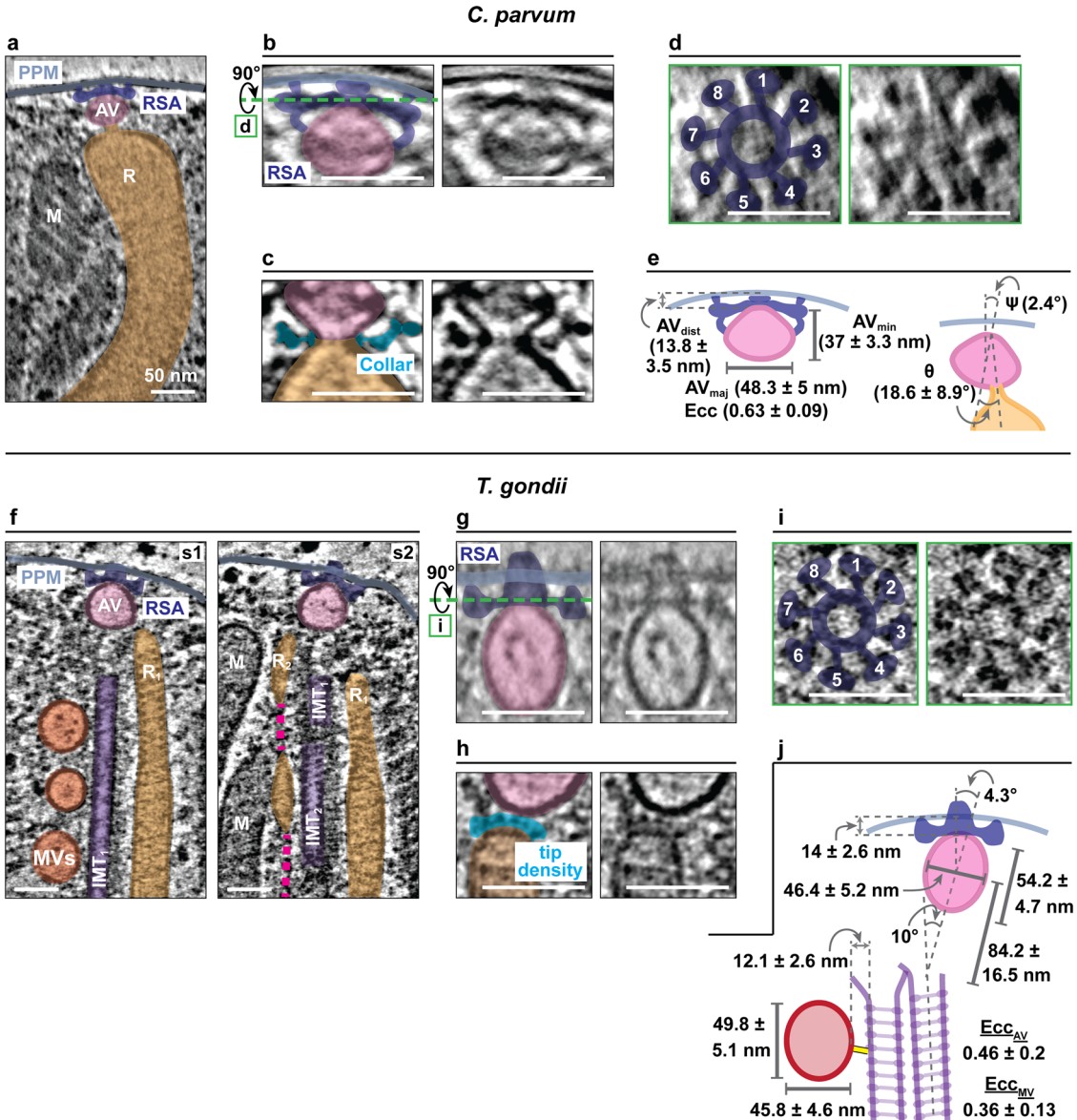

**Fig. 2 An apical vesicle (AV) and rhoptry secretory apparatus (RSA) together mediate docking of the rhoptry to plasma membrane.** Arrangement of rhoptry secretion system in *C. parvum* (**a**–**e**) and in *T. gondii* (**f**–**j**). **a**, **f** An apical vesicle (AV; pink) and rhoptry secretory apparatus (RSA; dark blue) connect rhoptries (R, $R_1$, $R_2$; orange) to the parasite plasma membrane (PPM; light blue) at the apical end. M: micronemes, IMT: intraconoidal microtubule (purple), and MVs: microtubule-associated vesicles (red). Rhoptry constrictions are denoted by a dark pink colored dashed line. **a** One rhoptry is fused with the AV in *C. parvum* while **f** two rhoptries are in close contact with the AV in *T. gondii* (s1 and s2 indicate two different sections of the same tomogram). **b**, **g** RSA mediates connection of the AV to the PPM. **c**, **h** AV-rhoptry junction in *C. parvum* and *T. gondii*—**c** rhoptry connects with the AV, forming a neck with a continuous membrane, which is decorated on the outside by collar-like densities (cyan) in *C. parvum* while **h** *T. gondii* shows docking (no fusion) of rhoptry with the AV, mediated by a tip density (cyan). **d**, **i** Top views of RSAs (from orthogonal sectioning planes labeled with green dashed lines in **b** and **g**, respectively) that show an 8-fold rotational symmetry. **e**, **j** Shape, size and anchoring parameters of AVs in *C. parvum* (**e**) and *T. gondii* (**j**); the latter also shows AV's arrangement with respect to IMTs (purple) and MV (red) parameters. $AV_{dist}$: shortest distance between the AV membrane and the plasma membrane; $AV_{maj}$: length of the AV major axis; $AV_{min}$: length of the AV minor axis; Ecc: eccentricity; Ψ: anchoring angle of the AV to the plasma membrane; θ: orientation of the AV neck with respect to the minor axis of AV. Measurements are presented as mean ± standard deviation or, in the case of Ψ, simply median. Source data for this figure are provided as a Source Data file. Scale bars in all panels are 50 nm.

rhoptry docking similar to the AV. These observations led us to a working model in which MVs could serve as new AVs in subsequent rounds of rhoptry secretion by moving along the IMTs and positioning at the apical tip. Consistent with this proposed role for IMTs and MVs in rhoptry secretion, their presence has been previously demonstrated in apicomplexans possessing >2 rhoptries (such as *Besnoitia*[30], *Babesiosoma*[39], *Dactylosoma*[40], and *Globidium*[39,41,42]) but shown to be lacking in *Cryptosporidium*[17] or *Plasmodium*[32,43], which possess a single

docked organelle or organelle pair, respectively. Additional observations from our tomograms are also consistent with our working model. We found the length of IMTs and their MV occupancy to be correlated (Supplementary Figs. 8e and 9) and the positioning and spacing of MVs to be regular (Supplementary Figs. 8f and 9). The most apical MV lined up close to the apical tip of the IMT ($L_1 = 29 ± 17$ nm with a 95% C.I. of 23–35 nm; Supplementary Fig. 8f) followed by regularly spaced MVs behind it. The posterior region of the IMT unoccupied by MVs ($L_{last}$) was

the most variable (79 ± 38 nm with a 95% C.I. of 65–93 nm; considerably larger in comparison to $L_1$ but not exceeding the required space for 2 MVs). Overall, this may suggest successive loading of MVs from the posterior and their tight packing towards the anterior tip, ready for loading AVs during successive secretion events. Consistent with this idea, we found that the AV in each *T. gondii* cell was uniformly positioned in front of the IMT anterior tip and at a regular distance, likely mediated by a cloud of amorphous material sandwiched between the two (Supplementary Fig. 10). With regards to MVs moving along IMT tracks, several lines of evidence suggest the likely involvement of cytoskeletal motor proteins. Our tomograms revealed that the distances between the MVs and their associating IMTs were uniform (12.1 ± 2.6 nm with a 95% C.I. of 11.5–12.7 nm; Supplementary Fig. 7f–h) and similar to the size of eukaryotic kinesins and dyneins[44–46] in addition to displaying linker-like densities between the two. Moreover, Dynein Light Chain 8a was recently shown to localize to the conoidal region of *T. gondii* and implicated in rhoptry docking at the apical end[47]. Overall, our working model ties MVs and IMTs to sequential rhoptry secretion based on detailed/quantitative structural analyses. Nonetheless, static images are limited in their ability to predict dynamic processes and further studies using molecular markers for MVs are required to fully test this model. The precise role of motor proteins and/or IMT dynamics in MV transport also remains to be fully tested.

**The rhoptry secretory apparatus anchors the AV.** All tomograms of *C parvum* and *T. gondii* revealed an apical rosette along with other proteinaceous densities that link the AV to the plasma membrane ($n > 150$ and >100, respectively, Fig. 2b, g); we refer to this entire feature as the rhoptry secretory apparatus (RSA). Top views of the RSA displayed a prominent eight-fold rotational symmetry (Fig. 2d, i and Supplementary Fig. 11a, b). To test the importance of the RSA in anchoring the AV, we used cryo-ET to image a *T. gondii* strain in which we conditionally knocked down *nd9* (*nd9-iKD*), a gene previously shown to be important for rosette formation and rhoptry secretion[26]. Homologs of this protein participate in the formation of an exocytosis-related surface rosette structure in ciliates[48,49], suggesting an overall conservation of protein components for exocytosis in the infrakingdom Alveolata. Tomograms of *nd9-iKD* cells showed variable RSA ultrastructure that displayed diminished contact between the AV and the plasma membrane, as well as the loss of 8-fold symmetry (Supplementary Fig. 11c); the latter is readily appreciated using subtomogram averaging and harmonic analysis (Supplementary Fig. 11d, e). Importantly, the mutant showed improper AV-anchoring (Supplementary Fig. 11c, f–h) while the AV morphology remained unaffected (Supplementary Fig. 11i). This finding links the RSA to previous extensive molecular studies[26] and support a model in which RSA and RSA-mediated AV anchoring to the plasma membrane are important for rhoptry secretion.

**Elaborate multi-component ultrastructure of the RSA.** In order to reveal structural details of the RSA, we performed subtomogram averaging on RSAs in both *C. parvum* and *T. gondii* (see Methods and Supplementary Figs. 12 and 13 for detailed procedure). The resulting averages revealed RSAs to be of remarkable complexity (Fig. 3). The apical rosette was clearly resolved along with additional densities on the cytoplasmic side extending towards and tightly interacting with the AV. The vertical and horizontal sections best illustrate the individual components of this apparatus (Fig. 3a–c, e–g), but the overall organization is better understood from the segmented volume

(Fig. 3d, h and Supplementary Movies 4 and 5). In both RSAs, the rosette was exposed to the cell exterior (diameter of 77 nm in *C. parvum* vs 63 nm in *T. gondii*; Fig. 3c(i), g(i)) with a central density (red) that resembled a pore. A feature with radiating spokes was found in close proximity to the central density on the cytoplasmic side (yellow; Fig. 3c(ii), d(iv), g(ii), h(iii, v)). This feature displayed a more prominent central region in *C. parvum* than in *T. gondii* (Fig. 3c(ii), g(ii)). It is currently unclear whether this difference represents functional states of a passage opening and closing (perhaps serving as a valve) or simply a structural difference between the two species. In both organisms, the central density was easily distinguishable from the outer ring of 8 densities (anchor-I, orange), suggesting that they could be performing different functions; the former possibly establishes a transient pore related to secretion while the latter anchors the RSA to the plasma membrane. The individual densities of anchor-I directly contacted those of anchor-II (dark blue) on the cytoplasmic side, which extended all the way to interact with the AV (Fig. 3b, d(ii), f, h(ii)). Each of the arms of anchor-II was in turn connected to one end of a slender density termed anchor-III (light blue). The anchor-III arms extended further than those of anchor-II to contact the AV membrane at the other end. Together, anchors-II and -III peripherally contacted the AV, likely contributing to the rigid anchoring of the AV while preventing spontaneous membrane fusion. In *C. parvum*, extensive interactions of anchor-II with the membrane of the AV could impart its teardrop shape (Supplementary Fig. 14a). Also, both anchor-II and anchor-III were long and twisted, and they contacted the AV like a clamp. In comparison, the arms of anchors-II and -III in *T. gondii* were much shorter and with less extensive interactions with the AV membrane (Supplementary Fig. 14c), which likely led to the greater variation in the anchoring angle of the AV in *T. gondii* (Supplementary Fig. 6f, m). We observed densities in the lumen of *C. parvum* AV that formed a posterior central channel (dark green) and extended radiating arms towards the AV membrane to anchor itself in place (Fig. 3b(iii), c(v), d(iii, v)). The posterior channel was not observed in *T. gondii* (Fig. 3f(ii), g(v), h(iii, iv)); however, both organisms showed an anterior central channel (light green) between the AV and the plasma membrane. There were also tentative protein densities embedded in the AV membrane at the entrance of the anterior central channels (arrows in Supplementary Fig. 14b, d). Overall, the two apicomplexan RSA ultrastructures show a previously uncharacterized type of machinery that is appropriately positioned between two opposing membranes to enable protein discharge and represents a conserved eukaryotic secretory mechanism in the infrakingdom Alveolata (see Supplementary Discussion for possible mechanisms of RSA-mediated membrane fusion and secretion of rhoptry proteins based on structural and molecular analogy to a similar machinery in ciliates[50,51]).

In summary, cryo-ET imaging and comparisons of two important apicomplexan parasites revealed a highly complex and conserved structural organization underlying the tight regulation and species-specific adaptations of rhoptry secretion (Fig. 4). They provide a structural framework to understand and investigate how rhoptries are shaped, primed and regulated for secretion, and yield working models that can now be tested in the context of host invasion. Mapping proteins onto these in situ structures and studying their mutant forms[26,52–54] provides a road map for further investigation that may uncover secretion mechanisms and new ways to prevent or treat several of the most important infectious diseases.

## Methods

**Preparation of *C. parvum* sporozoites.** *C. parvum* oocysts, purchased from Bunch Grass Farm (Deary, Idaho, USA), were excysted as previously described[55] with

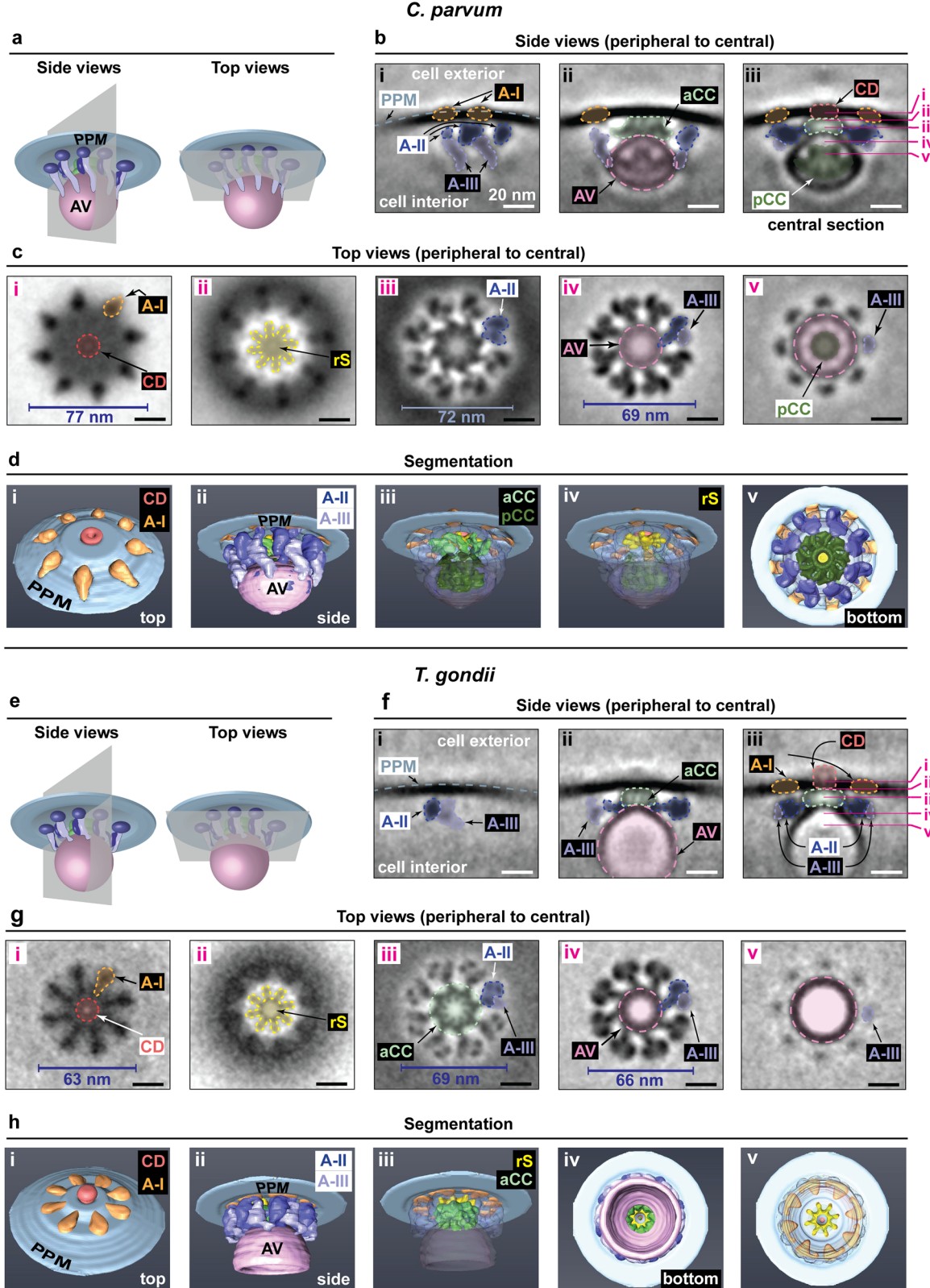

some modifications. Oocysts were washed 10 min in bleach at 4 °C and excystation was triggered by 0.8% sodium taurodeoxycholate (Sigma-Aldrich, St. Louis, MO, USA, Cat# T0875) for 10 min at 16 °C and subsequent incubation at 37 °C for 1 h. Parasites were then resuspended in PBS or DMEM-10 (consisting of DMEM—ThermoFisher, Waltham, MA, USA, Cat# 10313039; 10% FBS—Atlanta Biologicals, Flower Branch, Georgia, USA, Cat# S11550; L-Glutamine—Gemini Bio Products, CA, USA, Cat# 400-106; Penicillin–Streptomycin—ThermoFisher, Cat#

SV30010). 10 nm colloidal gold fiducials (Ted Pella, Redding, USA) were added to the suspension (for alignment purpose during tomogram reconstruction from tilt series). Approximately, 4 µl of suspended cells (estimated at ~4 × 10⁶ sporozoites) were applied onto Quantifoil 200 mesh copper R2/2 holey carbon EM grids, excess liquid blotted away, and plunge frozen in a liquid ethane/propane mixture (precooled with liquid nitrogen) using a EM GP2 automatic plunger (Leica Microsystems, Wetzlar, Germany)[56]. The blotting chamber was set to 95–100% relative

**Fig. 3 In situ ultrastructures of rhoptry secretory apparatus (RSA) from _C. parvum_ and _T. gondii_.** Subtomogram averages and 3-D displays of rhoptry secretory apparatus (RSA) in _C. parvum_ (**a–d**) and _T. gondii_ (**e–h**). **a, e** Schematic of the RSA, the apical vesicle (AV), and the parasite plasma membrane (PPM) demonstrating the side view and top view orientations used to generate the image slices in **b**, **c**, **f**, and **g**. Cross-section positions of each top view in **c** and **g** with respect to the side view in **b** and **f** are indicated by numbered lines in (**b**(iii)) and (**f**(iii)). **d, h** Perspective views of the 3-D segmentations of the subtomogram averages. Different levels of transparency for the outer layer of features reveal inner components (**d**(iii–iv), **h**(iii,v)). The RSAs are elaborate, multi-component structures spanning the AV and the PPM all the way to the cell exterior. They display a rotational 8-fold symmetry about a central vertical axis and can be structurally divided into individual components—the rosette containing anchor-I (A-I; orange) and central density (CD; red) extends into the plasma membrane to the cell exterior; anchor-II (A-II; dark blue) contacts anchor-I at one end and the AV membrane at the other; anchor-III (A-III; light blue) interacts with anchor-II at one end and with the AV membrane at the other; a posterior central channel (pCC; dark green) inside the AV (in _C. parvum_ alone) held in place by outward densities that connect with the AV membrane; an anterior central channel (aCC; light green) extends from the pCC towards the plasma membrane; and a feature with radiating spokes (rS; yellow) that contacts the CD. The feature with radiating spokes displayed a more prominent central region in _C. parvum_ compared to _T. gondii_. The putative pore of the CD seems sealed on the outside in _T. gondii_. The anchors-II and -III in _T. gondii_ seem to be shorter and making lesser contacts with the AV compared to those in _C. parvum_. Details of the subtomogram averaging schemes are provided in the Methods section and summarized in Supplementary Figs. 12 and 13. 3-D volumes of these final averages are provided in the repository referenced under "Data availability" for Supplementary Fig. 12. Source data for this figure are provided as a Source Data file. Scale bars in all panels are 20 nm.

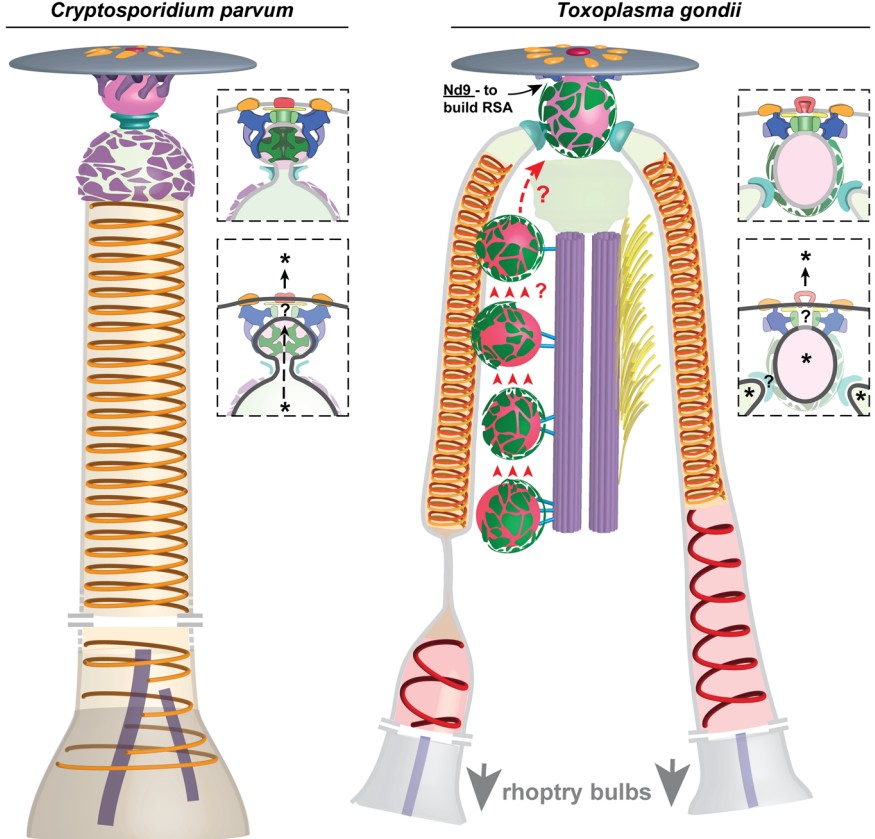

**Fig. 4 Models for the rhoptry secretion system in _C. parvum_ and _T. gondii_.** Salient findings of this study include conserved ultrastructures in _C. parvum_ and _T. gondii_ for—shaping the rhoptries (via helical filaments in the neck); priming rhoptries for secretion (via the apical vesicle (AV) and the rhoptry secretory apparatus (RSA) that dock rhoptries to the plasma membrane); and for regulating rhoptry secretion (via RSAs that likely perform controlled fusion of the AV with the plasma membrane in response to rhoptry secretion stimuli). In their resting state, the RSAs hold the AV away from the plasma membrane thus preventing spontaneous fusion (top insets). Likewise, the 'collar' proteins in _C. parvum_ and the rhoptry tip densities in _T. gondii_ likely regulate the fusion of rhoptries with the AV. The two fusion events together would complete the conduit for rhoptry proteins' secretion (denoted by * in bottom insets). _T. gondii_, in order to secrete multiple rhoptries (including the docked pair at the AV), likely requires multiple AVs. The microtubule-associated vesicles (MVs) are possibly trafficked to the cell apical tip (denoted by red arrows and arrow heads) to facilitate this function.

humidity at 37 °C and blotting was done from the sample side of the grid using Whatman filter paper #1. Plunge-frozen grids were subsequently loaded into autogrid c-clip rings (ThermoFisher). The autogrid containing frozen grids were stored in liquid nitrogen and maintained at ≤−170 °C throughout storage, transfer and cryo-ET imaging.

**Drug treatments for _C. parvum_.** After excystation (prior to plunge freezing), sporozoites resuspended in DMEM-10 were incubated at 37 °C with 5 μM A23187

(Sigma-Aldrich, Cat# C7522) for 4 min or with 1 μM Jasplakinolide (Sigma-Aldrich, Cat# J4580) for 40 min (or with 5 μM Jasplakinolide for 65 min).

**Preparation of _T. gondii_ tachyzoites.** _T. gondii_ tachyzoites from the RH strain (or the mutant _nd9-iKD_) were cultivated as previously described[54] with minor modifications. Tachyzoites were cultivated in monolayer of human foreskin fibroblasts (HFF – ATCC, CRL 1634) with DMEM-10 media supplemented with 5% fetal calf serum (FCS) and 2 mM glutamine. Upon egress, extracellular parasites were

collected and resuspended in PBS or DMEM-10 with 10 nm gold fiducials before loading ~$4 \times 10^6$ tachyzoite (in a 4 μl suspension) on to a grid. The nd9-iKD parasites were pre-treated with 1 μM ATc (Sigma-Aldrich, Cat# 37919) for 72 h before freezing.

**Cryo-electron tomography (cryo-ET).** Cryo-ET was performed on a Thermo-Fisher Krios G3i 300 keV field emission gun cryo-TEM. Dose-fractionated Imaging was performed using the SerialEM software[57] on a K3 direct electron detector[58] (Gatan Inc., Pleasanton, CA, USA) operated in the electron-counted mode. Motion correction of images was done using the Alignframe function in SerialEM. We additionally used the Volta phase plate[59,60] to increase contrast without high defocus, and the Gatan Imaging Filter (Gatan Inc., Pleasanton, CA, USA) with a slit width of 20 eV to increase contrast by removing inelastically scattered electrons[61]. After initially assessing cells at lower magnifications for suitability of ice thickness and plasma membrane integrity, tilt series were collected with a span of 120° (−60° to +60°; bi-directional scheme) with 2° increments at a magnification of 33,000X (with a corresponding pixel size of 2.65 Å) and a defoci range of −1 to −3 μm. The cumulative dose of each tilt-series ranged between 100 and 150 e−/Å². Once acquired, tilt series were aligned using the 10 nm colloidal gold as fiducials and reconstructed into tomograms by our in-house automated computation pipeline utilizing the IMOD software package[62].

**Quantifications, statistics and reproducibility.** We obtained a total of 285 tomograms for *C. parvum* (119 for untreated, 57 for A23187-treated, and 109 for Jasplakinolide-treated) and 100 for *T. gondii* from multiple frozen grids that were imaged over several sessions (at least 2 multiple-day sessions for each sample). Each of the following quantifications is from a subset of these tomograms that resolved the feature of interest (also randomly chosen if the number of tomograms exceeded the requirement). We note that parasites flattened on the grid, likely due to blotting. However, this flattening did not reflect on the shape of individual rhoptries, AV, MVs, or the RSAs. Flattening could have added noise to the relative positions of these features but their organizational patterns were evident despite the presence of such potential noise.

*Rhoptry measurements.* In the case of *C. parvum*, tomograms of 26 cells were analyzed (Supplementary Fig. 1) in which the filament-lined and non-filament-lined regions were clearly distinguishable. Measurements were made equidistant from each other every ~10 nm. Branch points were ignored. Similarly, for *T. gondii*, 31 docked rhoptries were analyzed from 19 different cells (Supplementary Fig. 2). Due to the limited field of view, tomograms of *T. gondii* did not capture the bulb and the part of the posterior rhoptry neck transition into the bulb.

The following measurements were made on 28 *C. parvum* cells and 25 wild-type *T. gondii* cells for AV-related measurements (Supplementary Fig. 6); 23 (Supplementary Fig. 7; except panel d, which used 22) and 28 (Supplementary Fig. 8) wild-type *T. gondii* cells for MV-related measurements; and 32 nd9-iKD *T. gondii* cells for AV related analyses (Supplementary Fig. 11).

*Rhoptry distance from AV.* Shortest distance between AV membrane and rhoptry tip membrane in *T. gondii*.

*θ.* Orientation of the AV neck with respect to the minor axis of AV (see below under 'vesicle dimensions' for more details) in *C. parvum*. Angle was measured between the minor axis of AV and the line connecting the rhoptry attachment point (on the AV membrane) to the farther end of the minor axis.

*Vesicle dimensions.* Each vesicle was described by approximating it to an ellipsoid and using only two axes (major and minor) instead of three for simplicity. In other words, one of the central sections—an approximated ellipse—was used to describe the vesicle. The longest axis for each vesicle in 3-D was marked as the major axis (labeled as $AV_{maj}$ or $MV_{maj}$) while the shortest axis orthogonal to the major axis and intersecting it at the centroid was marked the minor axis (labeled as $AV_{min}$ or $MV_{min}$).

*Eccentricity.* $(1-b^2/a^2)^{1/2}$, where 'a' is the semi-major axis, and 'b' is the semi-minor axis.

*$AV_{dist}$ (anchoring distance of AV).* Shortest distance between the AV membrane and parasite plasma membrane.

*Ψ (Orientation parameter for AV).* angle formed between the longitudinal axis of the AV (minor axis for *C. parvum* AV and major axis for *T. gondii* AV) and the normal to the parasite plasma membrane from the AV centroid.

*$MV_{dist}$.* Shortest distance between MV membrane and the closest IMT's surface.

*Closest IMT to MV.* Shortest distances from MV centroid to the central spines of the two IMTs were individually measured, and the shorter of the two decides the closest IMT.

*AV-IMT distance.* Distance between the AV centroid and the closest IMT tip.

*Φ.* Angle formed between the IMTs' trajectory and the line connecting the AV centroid to the closest IMT tip.

*Length of IMTs.* Length of IMTs were measured by placing contours that followed their trajectories.

*Length of IMT2 region associated with fibers ($L_f$).* The span of this region was measured on the IMT that was not associated with the MVs.

*Length of IMT occupied by MVs.* The individual diameters of the MVs were measured parallel to the IMT1 and the inter-MV distances were measured in a similar manner.

*MV positioning on IMTs.* Points were marked at IMT1 tips (anterior and posterior) and at points of attachment of MVs (points on IMT1 closest to each MV). Distances between these points were measured along the length of IMT1.

Quantifications were performed on IMOD generated models. All model files were exported for analysis and plotting using Numpy, Matplotlib and Seaborn libraries in Python 3.5. The corresponding graphs also report the mean ± standard deviation values. Alternatively, for skewed distributions that are not normal, we instead report the median value alone. In Supplementary Fig. 11f, g, *p*-values are reported for the differences between measurements in nd9-iKD cells and the corresponding measurements in wild-type cells using 2-sample Kolmogorov-Smirnov tests (ks_2samp method) in Scipy package of Python.

**Subtomogram averaging for the anterior neck filament of *T. gondii*.** We analyzed >25 tomograms of *T. gondii*; in all of them, we observed a helical geometry for the anterior neck filament with an ~11 nm pitch (11.1 ± 0.7 nm with a 95% C.I. of 10.9–11.3 nm; Supplementary Fig. 4 and Supplementary Movie 3). For subtomogram averaging, we chose 5 tomograms with superior signal-to-noise ratio and selected regions of the rhoptry anterior neck that were straight. We used IMOD to distribute model points along the central line of the rhoptry neck with spacing of 10 points per helical turn of the anterior neck filament. We then used cubic boxes of ~38 nm in side length to extract subtomograms around each model point. The boxes all have their Z-axis aligned to the line connecting the model points, but with a 36° rotation along the Z-axis on each successive model points. This way, a consistent phase for the helical filament was maintained and the densities were roughly aligned with each other. A total of 925 subtomograms were produced for alignment and averaging by Dynamo[63]. First, we produced an average for all the subtomograms with no search parameters to create a reference. We then applied a "hollow cylinder" mask to exclude the luminal and extramembraneous regions of the reference. The first alignment step (round 1) following this was performed on four-fold downsampled data and applied a relatively wide cone aperture search (30°) and an azimuth range equivalent to the arc length of a boxed filament (36°). Narrow shift limits were also applied. These were refined iteratively 7 times, each time using half the previous search range. The next two rounds were downsampled from unbinned data by a factor of 2 for alignment purpose. Each step used aligned subtomograms from the previous. The second round had a narrow cone sampling (3°) while the third round had no cone sampling. The parameters remained unaltered otherwise. The fourth and fifth rounds followed the same pattern as the second and third, but they were on fully unbinned data. The subtomogram averaging scheme along with the intermediate steps and results are illustrated in Supplementary Fig. 5a–g. An additional average was independently generated for the same filament using 1,100 subtomograms from another set of 7 tomograms and a similar workflow as the first average (Supplementary Fig. 5h). The second average was similar to the first, validating the generality of our observations including the helical geometry for the filament. Combining all the subtomograms from both averages did not further improve the filament details (Supplementary Fig. 5i). We have included the 3-D volumes for all these averages in the repository referenced under "Data availability".

**Subtomogram average for the collar in *C. parvum*.** A total of 41 subtomograms were averaged using PEET[64] without any symmetry.

**Evaluation of symmetry in nd9-iKD RSA versus wild-type RSA in *T. gondii***
*Subtomogram averaging.* Non-symmetrized averages of RSA were independently generated from wild-type *T. gondii* and nd9-iKD *T. gondii* using 25 unique RSA densities each and identical procedures using PEET. Representative tomogram sections for the two averages are shown in Supplementary Fig. 11d and the averages are included as 3-D volumes in the repository referenced under "Data availability".

*Harmonic analysis.* Rotational symmetry between 2- and 8-fold was assessed for individual RSA densities of the wild-type and nd9-iKD cells from representative 2-D sections (top views). Each RSA density was iteratively rotated by 360/fold-symmetry degrees from 0 to 360° (for example, for 8-fold symmetry, the angles will include 45°, 90°, 135°, 180°, 225°, 270°, 315°) starting from a randomized

starting orientation. This process was repeated using different RSA densities until approximately 20 images were obtained for each fold including the randomized starting positions. In other words, for accessing 2-fold symmetry, 10 unique RSA densities were each rotated by 180° to obtain 20 images in total while for accessing 8-fold symmetry, 3 unique densities were each iteratively rotated by 45° to obtain 24 images in total. Then, for each density, Pearson's correlation coefficients were calculated between the initial non-rotated (but randomized) image and each of the rotated images. To combine these results, all the correlation coefficients were finally normalized using the average value for 2-fold symmetry at 180° rotation. The results for the wild-type and nd9-iKD cells are shown in Supplementary Fig. 11e. Additionally, mean ± standard error values are reported for correlation coefficients corresponding to each angle. Graphing was performed using MATLAB.

### Generation of RSA subtomogram averages in *C. parvum* and *T. gondii*.

Alignment and averaging of subtomograms were performed using PEET. We manually inspected tomograms of both parasites at the apical end and identified RSA ultrastructures. Subtomogram boxes were centered in the middle of the rosette of intramembranous particles and manually rotated to the same orientation among RSA ultrastructures. In *C. parvum*, we initially performed subtomogram averaging on 39 RSAs without utilizing the rotational symmetry. Densities with 8-fold rotational symmetry were seen throughout the RSA in the average, from the extracellular region all the way to the AV (Supplementary Fig. 12a, b). We therefore utilized this symmetry during particle picking (thus yielding 8 times more subtomograms, increasing the total number to 312) to further enhance signal-to-noise ratio. The result was a better-resolved average that revealed the RSA to be an elaborate ensemble of components (Supplementary Fig. 12e, f). Previously, $Ca^{2+}$ signaling was shown to be involved in rosette-mediated exocytosis in ciliates[65], suggesting a possible role for $Ca^{2+}$ in apicomplexan rhoptry secretion. We therefore examined the RSA ultrastructures from *C. parvum* cells treated with A23187, a calcium ionophore that increases cytoplasmic $Ca^{2+}$ levels[65], and compare them with the ones from untreated cells. We observed no difference in the 8-fold symmetry of individual RSA densities and initial subtomogram averages between these two populations (Supplementary Fig. 12b, c) nor did we observe a discernable structural difference in their corresponding final subtomogram averages (344 subtomograms from 43 RSAs for A23187-treated cells; Supplementary Fig. 12b, e, f). We therefore combined the datasets from these two samples for subsequent structural analysis. We additionally included a dataset (376 subtomograms from 47 RSAs) from another drug treatment, Jasplakinolide, to further improve the resolution (Jasplakinolide, an actin stabilizer[66], was used to image the organism for an unrelated purpose, i.e., to study actin-based motility). Nonetheless, like A23187-treatment, Jasplakinolide-treatment did not result in structural change in the RSA (Supplementary Fig. 12b, d–f). Thus, we combined all the untreated and drug treated samples to obtain our final *C. parvum* RSA average that included a total of 129 unique RSAs (1032 subtomograms after incorporating the 8-fold symmetry). The workflow is summarized in Supplementary Fig. 13a. More details of the subtomogram averaging scheme are provided below and outlined in Supplementary Fig. 13b-h. For each *C. parvum* sample (untreated and two drug-treatments), the subtomograms were first averaged without any computational alignment to generate an initial reference. This reference was then applied with a spherical mask covering the AV, RSA and part of the plasma membrane and used to align all subtomograms. The alignment procedure started with narrow search ranges and search steps in all three Euler angles, and refined them further in 3 or 4 successive iterations. After each iteration, a new reference was generated using the best two-thirds of aligned subtomograms for the next iteration. We found that small angular searches were adequate because of careful manual particle picking at the beginning (resulting averages provided in Supplementary Fig. 12b). We then incorporated the 8-fold rotational symmetry of the RSAs by picking 8 subtomograms from each RSA particle (by iteratively rotating each particle by 45°) and repeated the same procedure for alignments, yielding the final average for each kind of *C. parvum* RSA (untreated, A23187-treated, Jasplakinolide-treated; Supplementary Fig. 12e, f). A similar workflow was implemented to generate the combined overall averages for the *C. parvum* RSA and *T. gondii* RSA as well (Fig. 3 and Supplementary Movies 4 and 5). For *T. gondii*, 66 unique RSA particles from untreated cells yielded 528 subtomograms after incorporating the 8-fold symmetry. The following subtomogram averages are provided (as 3-D volumes) in the online repository referenced under "Data availability" for Supplementary Fig. 12: (1) *C. parvum* RSA averages generated for each drug treatment (and untreated) without exploiting 8-fold symmetry; (2) *C. parvum* RSA average generated after combining all treatments but without exploiting 8-fold symmetry; (3) *T. gondii* RSA average generated without exploiting 8-fold symmetry; and (4) Final RSA averages generated for *C. parvum* and *T. gondii* after utilizing their 8-fold symmetry during particle picking (corresponding to the 2-D images provided in Fig. 3). Of the two final averages provided for *T. gondii*, "ToxoRSA_Fig3_final.mrc" contributed to top views and "ToxoRSA_Fig3_final_P416.mrc" side views in Fig. 3. The latter average includes only 416 good subtomograms to resolve some of the features a little better but its resolution was very similar to that of the former, which used 528 subtomograms. Fourier shell correlation plots (Supplementary Fig. 12g, h) were calculated using PEET.

### Figures presentation, modeling and segmentations.
Tomograms were favorably oriented in 3-D using IMOD's slicer window such that the desired tomogram section was in view for presentation. To enhance contrast, we averaged a few sections around this section of interest. Figures were prepared using Adobe Illustrator CC (Adobe Inc., San Jose, USA). Rhoptry filaments in Fig. 1 and Supplementary Figs. 1 and 2 were modeled using IMOD. Manual segmentations and subsequent animations for apical ends of *C. parvum* and *T. gondii*, and their RSA subtomogram averages in Supplementary Movies 1, 2, 4, and 5 were generated using Amira (ThermoFisher) and using IMOD for Supplementary Movie 3 and Fig. 1h.

**Reporting summary**. Further information on research design is available in the Nature Research Reporting Summary linked to this article.

## Data availability
Additional data generated in this study are provided in the Supplementary Information and Source Data file. Source data include tomogram sections without any color annotations. Three-dimensional tomogram volumes are also made available through an online repository – https://figshare.com/projects/In_situ_ultrastructures_of_two_evolutionarily_distant_apicomplexan_rhoptry_secretion_systems/112917; individual DOI links include https://doi.org/10.6084/m9.figshare.14527794.v3 (for Supplementary Fig. 2), https://doi.org/10.6084/m9.figshare.14538339.v5 (for Supplementary Fig. 5), https://doi.org/10.6084/m9.figshare.14534109.v2 (for Supplementary Fig. 6), https://doi.org/10.6084/m9.figshare.14527890.v2 (for Supplementary Fig. 11), and https://doi.org/10.6084/m9.figshare.14527911.v4 (for Supplementary Fig. 12). Any remaining minor datasets and analyses generated for the current study are available from the corresponding author upon request. Source data are provided with this paper.

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

## Acknowledgements

We thank Darrah Johnson for her technical assistance with the Krios G3i cryogenic electron microscope; the Singh Center for Nanotechnology and the Beckman Center for Cryogenic Electron Microscopy at the University of Pennsylvania for hosting and supporting the use of the Titan Krios; Eusha Hasan and Cameron Thompson for preliminary analyses of tomograms; Catherine Oikonomou for critical reading of the manuscript; and other members of the labs of Y.-W.C., B.S. and M.L. for their support and discussion. This work was supported in part by a David and Lucile Packard Fellowship for Science and Engineering (2019-69645) and a Pennsylvania Department of Health FY19 Health Research Formula Fund to Y.-W.C.; by National Institutes of Health grants R01AI112427 and R01AI127798 to B.S.; by European Research Council advanced grant 833309 (KissAndSpitRhoptry) to M.L.; by the Roy and Diana Vagelos Scholars Program in Molecular Life Sciences to support L.M.T.; by the Mary L. and Matthew S. Santirocco College Alumni Society Undergraduate Research Grant to W.D.C.; and by EMBO fellowship ALTF 58-2018 to A.G. M.L. is an INSERM researcher.

## Author contributions

S.K.M., A.G., B.S. and Y.-W.C. conceptualized and designed the experiments. A.G. cultured and isolated the parasites provided by M.L. and B.S. S.K.M. and A.G. prepared frozen grids and S.K.M. performed cryo-ET using an automated data-processing pipeline for rapid tomogram reconstruction that was established by W.D.C., who also provided

additional computational support during data collection, processing and management. S.K.M. analyzed the tomograms along with M.M., L.M.T. and Y.-W.C.; the latter three performed subtomogram averaging with input from S.K.M. L.M.T. performed harmonic analysis for assessing rotational symmetry of the RSA. A.G., M.L. and B.S. provided critical insights for interpretation of tomograms. S.K.M. and Y.-W.C. prepared the manuscript with contributions from A.G., L.M.T., M.L. and B.S.

## Competing interests

The authors declare no competing interests.
