## [Peer Review File · Nature Communications]

Reviewers' Comments:

Reviewer #1:

Remarks to the Author:

The authors have sought to characterise the molecular structures associated with rhoptries at the point of interaction and secretion. Rhoptries are a key organelle of apicomplexan parasites required for invasion of their hosts, and the subsequent infection and pathologies that they cause. The authors have chosen two distantly related parasites, both of human medical relevance, to draw overarching conclusions that might be relevant for many or even most apicomplexan parasites. Indeed, they have previously identified that parts of this apparatus are likely shared by more distant non-parasites such as ciliates. The authors have employed cryoET as a powerful way to investigate these structures riding on the spectacular improvements of resolution and biological relevance that this method provides due to the minimal processing of the sample from live cells.

The outcomes of this work are spectacular. The data presented now allow mechanistic interpretation to many previous observations of rhoptry function, ultrastructure, and behaviour, that were otherwise poorly understood for lack of these insights. These include the morphology and compositional segregation of the rhoptries that are key to their multistage function in early and later stages of invasion. Moreover, the identification and characterisation of the rhoptry secretion apparatus provides a clear framework for now investigating the regulation of secretion of this organelle as well as how its cargo might be delivered across 3 membranes into the host cytoplasm.

I've no doubt that this study will reshape thinking and research into invasion in these important parasites, and this work is equally relevant to Plasmodium. I think it is a small shame that elements of the discussion are separated into a Supplementary Discussion as these are very relevant and logical to the study. But this is a journal issue, not a criticism of the work or report. Otherwise, I want to congratulate the authors for their achievements here.

Minor issues to be considered are as follows:

Line 55: I wouldn't say that the proteins have to 'traverse' three membranes to get to the host cytosol as the mechanisms might include membrane fusion and opening, and thus no need for traversal per se. Perhaps better to say that these proteins have to be relocated across these three membranes.

Line 81: I think it would help to describe what two- versus one-start helices are. The differences are not very obvious from the figures.

The evidence for treadmilling of the IMTs and this being a mechanism for AV delivery is very speculative. The splayed MT ends could be stabilized in this state. I suggest being more cautious about linking these multiple unknowns into a likely hypothesis.

Line 171: 'in the ballpark' is an Americanism that is not necessarily interpretable for all readers.

Line 191: Alveolata has the taxonomic rank of Infrakingdom, not superphylum.

line 203: what was the basis of the anticipated complexity of the RSA that was not met by the data? Without context of what was 'anticipated' it is a little hard to interpret the authors' reaction here. Minor point, but something to consider.

The rationale for Jasplakinolide treatment is not stated. In Toxoplasma it can cause a major structural change to the apical complex. Some description of why it has been used for Cryosporidium, and what the expected effects are, is warranted.

Reviewer #2:
Remarks to the Author:
Remarks to the Authors

Although rhoptries play a major role in host-cell invasion by Apicomplexan parasites, many aspects of the structure and functions of these important secretory organelles remain poorly understood in the absence of structural information at resolutions better than 5-10nm. This manuscript seeks to use cryoET of two Apicomplexan parasites to address the fundamental questions of how the distinctive shape of the rhoptry is maintained, how proteins are segregated in the neck versus the bulb of the rhoptry, and what the mechanism is by which proteins are secreted from the rhoptries into the host cell, all of which are long-standing mysteries in the field. Unfortunately, the data presented is not sufficient to substantiate the claims laid forth in this study, and the basic premise of the manuscript is not justified by the data.

1. Line 2: The definitive title is misleading and should be adjusted to reflect the fact that only two species out of the large, diverse Apicomplexan phylum are represented in this work, especially given that the two models presented here exhibit distinct differences. Furthermore, at resolutions of 4-5nm, without knowledge of the identities of any of the proteins present, the symmetrical arrangements of proteins presented here cannot justifiably be called structures. Ultrastructure may be a more appropriate descriptor.
2. Lines 78-83, lines 343-361: It is not clear from the description provided how it was determined that the putative rhoptry neck filaments exhibit helical symmetry, nor how the helical symmetry was determined.
3. Line 89-91, 95-97: Figure 1: In the raw images from tomogram slices, an apparent helical ultrastructure can be seen for the posterior filament, this is not the case for the anterior filament. The raw image only shows dot shaped densities lining the interior of the neck. The only images included are subtomogram averages, but no convincing raw, unprocessed data supporting a helical structure is shown. The raw tomogram including all the slices should be provided.
4. Line 92-95, lines 343-361: A more detailed description of the methods used for subtomogram averaging of the putative filaments should be provided, with a figure that includes a comprehensive processing scheme showing how the boxes were chosen, all the intermediate processing steps, and the results of each intermediate step. Additionally, the potential for biased processing and introduction of artifacts needs to be addressed by showing all steps processed with and without symmetry and masking.
5. Lines 102-104, 106-108, 111-112, 137-138, 159-161, 167-172, 177-178, 233-237: These are speculative statements that would be more appropriate in the context of a discussion rather than results.
6. Lines 137-138: No data is presented to support the speculation that the AV coordinates secretion from a pair of rhoptries in *T. gondii*.
7. Lines 159-163: The observation that the AV is consistently positioned the same distance from the front of the IMT tip is a single "snapshot", and is not sufficient to support the speculation that the MVs travel along the IMTs and replace the AV positioned at the apex of the parasite. Time-lapse data of some sort showing the MVs moving along the IMTs and then replacing the AV would be needed.
8. Lines 163-172: This is an overinterpretation of the data presented. The data shown is a snapshot in time, which does not allow for conclusions beyond qualitative description of a single state. The speculation about the dynamics of the IMTs and/or the potential role of dyneins/kinesins in transporting MVs belongs in a discussion section, not in the results.
9. Lines 233-237: There is no data presented to support the speculation that secreted proteins passes through the RSA.
10. Lines 241-242: The data presented here are not sufficient to support the claims set forth in this statement.
11. Lines 182, 241, 244: "ultrastructure(s)" would be a more appropriate word to use here, rather than "structure(s)".

12. Lines 300, 303: A larger number of tomograms, drawn from a larger number of cells, would be needed to justify the claim that these observations are general within *C. parvum*.
13. Line 344: Why were so few tomograms used for this analysis? A significantly larger number of rhoptry necks, drawn from a larger number of tomograms, would be needed to justify the claim that these observations are general within *T. gondii*. Furthermore, the resolution achieved here is not sufficient to support the claim that these putative filaments are indeed helical in nature.
14. Lines 364-366: For the subtomographic reconstruction of the RSA, data showing the initial C1 reconstruction should also be included.
15. Lines 384-420: complete schema detailing all of the intermediate steps (and corresponding results) in the data processing workflows for each of these reconstructions should be included.
16. Lines 396-407: The separate initial reconstructions of each group (no drug, Jasplakinolide, and A23187) should be included.
17. In all figures, for each of the denoised tomogram sections shown, the original aligned tomograms should be provided.
18. Supplementary Videos 1 & 2: In the 360 rotating view (00:23 – 00:30 in both videos), the reconstructions look strangely flattened – almost rectangular. Do the apical ends of *C. parvum* sporozoites and *T. gondii* tachyzoites normally form a flattened tip, like the sharp end of a spade, or do they come to a rounded point, like the end of a pencil? If the latter, it is possible that there has been some compression of the tips of the parasites during vitrification, perhaps due to the surface tension from the buffer after blotting. This could affect many of the distance measurements made in the manuscript, and should be acknowledged in the main text.
19. Extended Data Figure 3: Please provide the tomogram and reconstruction, just the single image is not sufficient – please also show the full tomographic section(s) from which these cropped images are drawn, and indicate on the full tomographic section(s) where the cropped images are drawn from. Given the relatively small sample size, caution needs to be taken in using these distance measurements for mechanistic conclusions (line 128), especially given the previously stated concerns that there may be some compression artifacts from the freezing process.
20. Extended Data Figure 4: “MVs were sometimes associated with free rhoptries, illustrating their compatibility for rhoptry docking...” How often is “sometimes?”
21. Is the RSA present in 100% of the parasites imaged?
22. Are there enough particles of the RSA to see if there are two species – open and closed? Could a 3D classification be done to determine whether multiple species/conformations are present?
23. Given that these parasites are not in contact with a host cell, why would the RSA plug be open in some instances?
24. Figure 3 is less informative than the Supplemental movies.
25. Figure 4: Parts of this model are not sufficiently substantiated with data. It needs to be more clearly stated that much of this data comes from a small number of tomograms, which do not justify universal claims.
26. The question of how proteins are organized inside the rhoptries is mentioned in the introduction/abstract/etc, but is not addressed by the data presented. Only a speculation on proteins possibly binding the helical filaments in the neck is given, but there is no experiment presented aimed at investigating this hypothesis.
27. In many of the points presented above, raw tomograms need to be provided. Furthermore, larger sample sizes would need to be provided for any claims of a general mechanism common across the species.

Reviewer #3:

Remarks to the Author:

Mageswaran et al. report on cryo-electron tomography studies of the rhoptry secretion complex of the apicomplexan parasites *Cryptosporidium parvum* and *Toxoplasma gondii*. This study delivers new structural insights into this complex including filaments lining the lumen of the rhoptry, and a rhoptry secretion apparatus with components on the internal and external face of the parasites apical tip. The

study provides fine structural detail on more well-known structures of the complex including the apical vesicles (AVs), intraconoidal microtubules and their potential role in trafficking additional AVs to the apical tip in *T. gondii*. There is a high level of new structural detail provided in the study that help to unravel how the apicomplexan rhoptry secretory system works. They use a conditional knock-down of nD9, a protein associated with previously reported external membrane apical rosettes, to confirm that the RSA is associated with the apical rosettes and has a role in rhoptry secretion. The manuscript is well written and a pleasure to read. The imaging, analysis and models proposed are detailed with the data used to be build the models coming from repeat measures across the study. This is a really nice study that will lead to new and exciting apicomplexan invasion biology findings in the future.

Minor Comments:

-Please state the organism for the images in Extended Data Fig 6 and 10.

REVIEWER COMMENTS

Reviewer #1 (Remarks to the Author):

The authors have sought to characterise the molecular structures associated with rhoptries at the point of interaction and secretion. Rhoptries are a key organelle of apicomplexan parasites required for invasion of their hosts, and the subsequent infection and pathologies that they cause. The authors have chosen two distantly related parasites, both of human medical relevance, to draw overarching conclusions that might be relevant for many or even most apicomplexan parasites. Indeed, they have previously identified that parts of this apparatus are likely shared by more distant non-parasites such as ciliates. The authors have employed cryoET as a powerful way to investigate these structures riding on the spectacular improvements of resolution and biological relevance that this method provides due to the minimal processing of the sample from live cells.

The outcomes of this work are spectacular. The data presented now allow mechanistic interpretation to many previous observations of rhoptry function, ultrastructure, and behaviour, that were otherwise poorly understood for lack of these insights. These include the morphology and compositional segregation of the rhoptries that are key to their multistage function in early and later stages of invasion. Moreover, the identification and characterisation of the rhoptry secretion apparatus provides a clear framework for now investigating the regulation of secretion of this organelle as well as how its cargo might be delivered across 3 membranes into the host cytoplasm.

I've no doubt that this study will reshape thinking and research into invasion in these important parasites, and this work is equally relevant to Plasmodium. I think it is a small shame that elements of the discussion are separated into a Supplementary Discussion as these are very relevant and logical to the study. But this is a journal issue, not a criticism of the work or report. Otherwise, I want to congratulate the authors for their achievements here.

We are grateful for the reviewer's many kind comments on the work and we share their enthusiasm for the insights gained into an organelle that is at the center of apicomplexan intracellular parasitism.

Minor issues to be considered are as follows:

Line 55: I wouldn't say that the proteins have to 'traverse' three membranes to get to the host cytosol as the mechanisms might include membrane fusion and opening, and thus no need for traversal per se. Perhaps better to say that these proteins have to be relocated across these three membranes.

We have changed the wordings as suggested. Line 53 now reads – "*The proteins therefore have to be relocated across three membranes*".

Line 81: I think it would help to describe what two- versus one-start helices are. The differences are not very obvious from the figures.

We now describe the two conformations (in Line 115) as follows – "*In C. parvum, we noted two conformations, a two-start (Fig. 1c) and a one-start helix (Extended Data Fig. 1d), both right-handed; while the latter is constituted by a single linear filament, the former is composed of two such filaments – shown in orange and blue – that alternately contribute to the helical turns.*"

The evidence for treadmilling of the IMTs and this being a mechanism for AV delivery is very

speculative. The splayed MT ends could be stabilized in this state. I suggest being more cautious about linking these multiple unknowns into a likely hypothesis.

We have now removed treadmilling of the IMTs as a mechanism for AV delivery from our discussion and instead kept the implications of IMT dynamics open ended. Line 233 now reads – *“The precise role of motor proteins and/or IMT dynamics in MV transport also remains to be fully tested.”*

We further performed an extensive analysis on these fibrous densities (n=28 cells). We confirmed that the two intraconoidal microtubules (IMTs) are complete and the fibrous densities form an additional putative microtubule (Extended Data Figure 5d). Moreover, the length of the IMT that associates with this feature is highly variable. Accordingly,

Line 200 now reads – *“We noted fibrous densities associated with IMTs ”*

Line 205 in Supplementary Information reads – *“MVs (red) are arranged on one of the IMTs (IMT1; both IMTs in purple) while the other IMT (IMT2) closely interacts with a cloud of fibrous material (yellow) that forms a putative partially-assembled microtubule (in addition to the two whole IMTs). Inset: A plot showing that the length of IMT region associated with the fibrous material (on IMT2) is highly variable.”*

Line 171: 'in the ballpark' is an Americanism that is not necessarily interpretable for all readers.

In Line 227, we have now changed “in the ballpark” to “similar to the size of”.

Line 191: Alveolata has the taxonomic rank of Infrakingdom, not superphylum.

Thank you for the correction. We have changed “superphylum” to “infrakingdom” throughout the manuscript

line 203: what was the basis of the anticipated complexity of the RSA that was not met by the data? Without context of what was 'anticipated' it is a little hard to interpret the authors' reaction here. Minor point, but something to consider.

We anticipated a membrane fusion machinery of the size and complexity of SNAREpins^{1,2}. What we ended up seeing was a more elaborate structure with cytosolic, membrane-associated (with the apical vesicle and plasma membrane) and extracellular components. We have changed “unanticipated” to “remarkable”. Line 258 now reads – *“The resulting averages revealed RSAs to be of remarkable complexity.”*

The rationale for Jasplakinolide treatment is not stated. In Toxoplasma it can cause a major structural change to the apical complex. Some description of why it has been used for Cryptosporidium, and what the expected effects are, is warranted.

We used Jasplakinolide treatment for an unrelated purpose, i.e., to study the actin-mediated motility in Cryptosporidium. We checked if the drug disrupted the RSA structure by independently averaging and comparing this average to that of untreated cells. We found that the overall arrangement of RSA was preserved and therefore included the subtomograms from this drug treated sample as well into our final RSA average to further improve signal-to-noise. Line 491 under the subtitle “Generation of RSA subtomogram averages in *C. parvum* and *T. gondii*” of the Methods section now reads – *“We additionally included a dataset (376 subtomograms from 47 RSAs) from another drug treatment, Jasplakinolide, to further improve the resolution (Jasplakinolide, an actin stabilizer was used to image the organism for an unrelated purpose, i.e., to study actin-based motility). Nonetheless, like A23187-treatment, Jasplakinolide-treatment did not result in structural change in the RSA (Extended Data Fig. 9b, d-f). Thus, we combined all the untreated and drug treated samples to obtain our final *C. parvum* RSA average that included a total of 129 unique RSAs (1,032 subtomograms after incorporating the 8-fold symmetry).”*

Reviewer #2 (Remarks to the Author):

Remarks to the Authors

Although rhoptries play a major role in host-cell invasion by Apicomplexan parasites, many aspects of the structure and functions of these important secretory organelles remain poorly understood in the absence of structural information at resolutions better than 5-10nm. This manuscript seeks to use cryoET of two Apicomplexan parasites to address the fundamental questions of how the distinctive shape of the rhoptry is maintained, how proteins are segregated in the neck versus the bulb of the rhoptry, and what the mechanism is by which proteins are secreted from the rhoptries into the host cell, all of which are long-standing mysteries in the field. Unfortunately, the data presented is not sufficient to substantiate the claims laid forth in this study, and the basic premise of the manuscript is not justified by the data.

We appreciate the reviewer's concerns and address them below.

1. Line 2: The definitive title is misleading and should be adjusted to reflect the fact that only two species out of the large, diverse Apicomplexan phylum are represented in this work, especially given that the two models presented here exhibit distinct differences. Furthermore, at resolutions of 4-5nm, without knowledge of the identities of any of the proteins present, the symmetrical arrangements of proteins presented here cannot justifiably be called structures. Ultrastructure may be a more appropriate descriptor.

We changed the title to "*In situ ultrastructures of two evolutionarily distant apicomplexan rhoptry secretion systems*". We changed "structure" to "ultrastructure" at relevant places throughout the manuscript.

2. Lines 78-83, lines 343-361: It is not clear from the description provided how it was determined that the putative rhoptry neck filaments exhibit helical symmetry, nor how the helical symmetry was determined.

We analyzed the rhoptry neck filaments in raw tomograms (> 25 tomograms for each organism) and found all of them to be helical in geometry. The anterior filaments of *T. gondii* showed the most regular helix with a pitch of ~11 nm (Supplementary Fig. 2 and Supplementary Video 3). We picked subtomograms along the length of rhoptries using this helical geometry to generate a subtomogram average for the anterior filament; in other words, 10 subtomograms were picked for every helical turn, each 1.1 nm away and rotated by 36° (360° for a full helical turn divided by 10 subtomograms) with respect to the closest subtomogram. This ensured the initial orientation of all subtomograms to be roughly the same. Their orientations were further refined (detailed in the new Supplementary Fig. 3) to improve their alignment and to correct for any artifacts from subtomogram extraction. The resulting average recapitulates the helical geometry with better signal-to-noise. We have now replaced the term "symmetry" with "geometry" and better clarified the subtomogram averaging procedure.

Line 112 – "*We found filaments lining the luminal surface of the rhoptry neck membrane in both C. parvum and T. gondii (n > 25 cells for each organism; Fig. 1b, f and Extended Data Figs. 1 and 2 show representative images; images without overlaid annotations are shown in Supplementary Fig. 6). Careful examination of raw tomograms in both organisms revealed their helical geometry.*"

Line 126 – "*In T. gondii, anterior and posterior neck filaments could be differentiated (from raw tomograms (Fig. 1f))*"

Line 416 – "*We analyzed > 25 tomograms of T. gondii; in all of them, we observed a helical geometry for the anterior neck filament with a 11 nm pitch (Supplementary Fig. 2 and Supplementary Video 3). For subtomogram averaging, we chose 5 tomograms with superior single-to-noise ratio and selected regions of the rhoptry anterior neck that were straight.*"

3. Line 89-91, 95-97: Figure 1: In the raw images from tomogram slices, an apparent helical ultrastructure can be seen for the posterior filament, this is not the case for the anterior filament. The raw image only shows dot shaped densities lining the interior of the neck. The only images included are subtomogram averages, but no convincing raw, unprocessed data supporting a helical structure is shown. The raw tomogram including all the slices should be provided.

We now provide top, middle and bottom raw tomogram slices for several examples of the *T. gondii* anterior rhoptry filament (Supplementary Fig. 2). Moreover, we have included a video (Supplementary Video 3) that shows raw tomographic sections through the anterior filamented regions from multiple cells along with 3-D segmentations of the helical filaments to support the observation.

4. Line 92-95, lines 343-361: A more detailed description of the methods used for subtomogram averaging of the putative filaments should be provided, with a figure that includes a comprehensive processing scheme showing how the boxes were chosen, all the intermediate processing steps, and the results of each intermediate step. Additionally, the potential for biased processing and introduction of artifacts needs to be addressed by showing all steps processed with and without symmetry and masking.

We now provide a detailed schematic for the subtomogram averaging procedure along with results for each of the intermediate steps (Supplementary Fig. 3). We note that we did not impose any symmetry during averaging. We simply used the geometry of the helical filament (estimated in raw tomograms) to pick subtomograms in a pre-aligned orientation. We have added the following information for more clarity. We also provide the 3-D volumes for these averages via an online repository (<https://doi.org/10.6084/m9.figshare.14538339.v4>).

Line 416 – “*We analyzed > 25 tomograms of T. gondii; in all of them, we observed a helical geometry for the anterior neck filament with a 11 nm pitch (Supplementary Fig. 2 and Supplementary Video 3). For subtomogram averaging, we chose 5 tomograms with superior single-to-noise ratio and selected regions of the rhoptry anterior neck that were straight.*”

5. Lines 102-104, 106-108, 111-112, 137-138, 159-161, 167-172, 177-178, 233-237: These are speculative statements that would be more appropriate in the context of a discussion rather than results.

We agree that these statements belong in the discussion section. The manuscript is formatted to combine Results and Discussions which makes this separation less obvious but can improve readability as suggested by reviewers #1 and #3). We hear the reviewer’s concern clearly and we now distinguish the presentation of data more clearly from interpretation and speculation. We have also indicated in the subheading that we have combined Results and Discussions

6. Lines 137-138: No data is presented to support the speculation that the AV coordinates secretion from a pair of rhoptries in *T. gondii*.

We show evidence for consistent AV coordinated *docking* of two rhoptries in *T. gondii*. While this arrangement is likely coordinating secretion of the two rhoptries as well, we agree that we do not formally demonstrate that. We therefore change line 177 – “*Altogether, the AV is a consistent component of the rhoptry system in these evolutionarily distant apicomplexans, and able to coordinate docking of a single rhoptry in C. parvum and utmost a pair of rhoptries in T. gondii.*”

7. Lines 159-163: The observation that the AV is consistently positioned the same distance from the front of the IMT tip is a single “snapshot”, and is not sufficient to support the speculation that the MVs travel along the IMTs and replace the AV positioned at the apex of the parasite. Time-

lapse data of some sort showing the MVs moving along the IMTs and then replacing the AV would be needed.

We agree that the data presented are snapshots of this postulated model. A true time-lapse experiment requires us to follow the position of individual MVs during invasion by live cell microscopy. Unfortunately, this is currently not feasible. We do not know the molecular composition of the MV and thus cannot target the structure with a suitable genetically encoded fluorescent tag. Furthermore, individual MVs are of a dimension not resolved by fluorescence microscopy and therefore not suitable for live observation. Observation of different stages of the process by cryo-ET may be feasible but rhoptry secretion only occurs in the context of host cell invasion. Realistically, such observations will depend on the isolation of novel mutants arrested during the act of secretion, and given the dimensions of the host cell will require additional technically challenging experiments such as fluorescence-guided focused ion beam milling. Ultimately, we are confident that such studies can be conducted, but they are not within the reach of the current manuscript.

To further test this idea within the boundaries of static observations, we performed a new set of careful quantitative analyses of the relative positioning of MVs on IMTs (n=28 cells). These produced interesting observations:

Line 212 – *“Additional observations from our tomograms are also consistent with our working model. We found the length of IMTs and their MV occupancy to be correlated (Supplementary Fig. 4 and Extended Data Fig. 5e) and the positioning and spacing of MVs to be regular (Extended Data Fig. 5f and Supplementary Fig. 4). The most apical MV lined up close to the apical tip of the IMT ($L_1 = 29 \pm 17$ nm; Extended Data Fig. 5f) followed by regularly spaced MVs behind it. The posterior region of the IMT unoccupied by MVs (L_{last}) was the most variable ranging from 35 nm to 150 nm (79 ± 38 nm; considerably larger in comparison to L_1 but not exceeding the required space for 2 MVs). Overall, this may suggest successive loading of MVs from the posterior and their tight packing towards the anterior tip, ready for loading AVs during successive secretion events.”*

To acknowledge the reviewer’s point, we now explicitly say that this is a working model. Line 206 – *“These observations lead us to a working model in which MVs could serve as new AVs in subsequent rounds of rhoptry secretion by moving along the IMTs and positioning at the apical tip.”*

Line 230 – *“Overall, our working model ties MVs and IMTs to sequential rhoptry secretion based on detailed/quantitative structural analyses. Nonetheless, static images are limited in their ability to predict dynamic processes and further studies using molecular markers for MVs are required to fully test this model. The precise role of motor proteins and/or IMT dynamics in MV transport also remains to be fully tested.”*

We believe that this working model will be a significant stimulus for further studies using genetic, biochemical and structural approaches in a range of organisms and this potential is evident in the enthusiasm of reviewers #1 and #3.

8. Lines 163-172: This is an overinterpretation of the data presented. The data shown is a snapshot in time, which does not allow for conclusions beyond qualitative description of a single state. The speculation about the dynamics of the IMTs and/or the potential role of dyneins/kinesins in transporting MVs belongs in a discussion section, not in the results.

We followed the reviewer’s suggestion and now state (line 233) – *“The precise role of motor proteins and/or IMT dynamics in MV transport also remains to be fully tested.”*

9. Lines 233-237: There is no data presented to support the speculation that secreted proteins passes through the RSA.

We agree that we have not captured the dynamic process of rhoptry protein secretion through the RSA in our cryo-ET experiments. Nonetheless, there is significant support for this working model. Rhoptry secretion occurs through the narrow apical tip of the zoite in a highly regulated fashion. We demonstrate the presence of an elaborate structure with eight-fold symmetry at this very position. We show RSA, AV and rhoptries are consistently arranged in a contiguous fashion in *Cryptosporidium* and *Toxoplasma*. Comparative genomics and comparative cell biology showed marked similarities in the overall structure (a rosette with eight-fold symmetry) and protein composition (Nd proteins) between the trichocyst of ciliates and the rhoptry of Apicomplexa³; rhoptry secretion is linked to Nd proteins and Nd proteins to the rosette in both ciliates and apicomplexans. We show that loss of Nd proteins impacts RSA structure. The overall conservation of structure and components makes conservation of their function very likely. We note that the most straightforward model for regulated rhoptry protein secretion is through the RSA (as shown in ciliates^{4,5}) and that no alternative model better supports this function. We have moderated our statements following the reviewer's suggestion.

Line 251 – *“This finding links the RSA to previous extensive molecular studies and support a model in which RSA and RSA-mediated AV anchoring to the plasma membrane are important for rhoptry secretion.”*

Line 290 – *“Overall, the two apicomplexan RSA ultrastructures show a previously uncharacterized type of machinery that is appropriately positioned between two opposing membranes to enable protein discharge and represents a conserved eukaryotic secretory mechanism in the infrakingdom Alveolata (see Supplementary Discussion for possible mechanisms of RSA-mediated membrane fusion and secretion of rhoptry proteins based on structural and molecular analogy to a similar machinery in ciliates).”*

10. Lines 241-242: The data presented here are not sufficient to support the claims set forth in this statement.

As suggested by the reviewer we modified the statement. Line 299 now reads – *“They provide a structural framework to understand and investigate how rhoptries are shaped, primed and regulated for secretion, and yield working models that can now be tested in the context of host invasion.”* We believe that the modified statement is well substantiated by the data presented. First, we provide evidence for rhoptry shaping by filaments with expanded datasets. Second, we show evidence for rhoptries being docked and primed for secretion at the plasma membrane via the RSA and the AV; in the absence of proper docking in the Nd9 depletion mutant (shown in this study), secretion is defective (shown in our previous study³). Third, to facilitate controlled secretion at the right time, i.e., during host cell invasion, the rhoptries are docked and ready, somewhat analogous to docking of synaptic vesicles in the neurons. The unprecedented level of detail for the RSAs – their extracellular, transmembrane and intracellular components – suggests a possible mechanism for coupling extracellular events such as host cell attachment to intracellular events such as membrane fusion to facilitate rhoptry secretion. We have also revealed a role for multiple membrane fusion events in rhoptry secretion, i.e., one between rhoptry and the AV (membrane fusion already shown for *C. parvum*) and likely another between the AV and the plasma membrane (based on molecular, structural and functional conservation between apicomplexans and ciliates), both of which represent regulation of rhoptry secretion.

11. Lines 182, 241, 244: “ultrastructure(s)” would be a more appropriate word to use here, rather than “structure(s)”.

We followed the reviewer's suggestion and now use the term ultrastructure(s).

12. Lines 300, 303: A larger number of tomograms, drawn from a larger number of cells, would be needed to justify the claim that these observations are general within *C. parvum*.

We previously showed analyses from a small sample size - 5 rhoptries (from 5 cells) for *C. parvum* and 6 rhoptries (from 3 cells) for *T. gondii*. We followed the reviewer's advice and significantly expanded the dataset to a total of 26 rhoptries (from 26 cells) for *C. parvum* and 31 rhoptries (from 19 cells) for *T. gondii*. This enhanced resolution and fully validated our conclusion.

Line 362 – “*In the case of C. parvum, tomograms of 26 cells were analyzed*”

Line 365 – “*Similarly, for T. gondii, 31 docked rhoptries were analyzed from 19 different cells.*”

13. Line 344: Why were so few tomograms used for this analysis? A significantly larger number of rhoptry necks, drawn from a larger number of tomograms, would be needed to justify the claim that these observations are general within *T. gondii*. Furthermore, the resolution achieved here is not sufficient to support the claim that these putative filaments are indeed helical in nature.

In fact, we observed a consistent helical architecture for the anterior filament in all > 25 tomograms analyzed in *T. gondii* (now with additional examples shown in Supplementary Fig. 2 and Supplementary Video 3). Due to the helical geometry of this filament, we were able to extract 925 subtomograms from 5 tomograms, enough for averaging to boost contrast and resolve a double stranded helical filament with membrane connections. We now generated an additional subtomogram average (using 1,100 subtomograms extracted from an independent dataset of 7 tomograms) that revealed a very similar filament to the first average (now shown in Supplementary Fig. 3h), thus supporting the generality of our observation. Combining all the subtomograms from both these averages did not improve the filament details any further (Supplementary Fig. 3i). We now provide 3-D volumes for all three averages via an online repository - <https://doi.org/10.6084/m9.figshare.14538339.v4>).

14. Lines 364-366: For the subtomographic reconstruction of the RSA, data showing the initial C1 reconstruction should also be included.

We had previously included representative slices from these non-symmetrized subtomogram averages in Extended Data Fig. 7d. As requested, we now provide entire 3-D volumes for the averages via an online repository (<https://doi.org/10.6084/m9.figshare.14527890.v1>). We now also point to it in the legend of Extended Data Fig. 7d.

15. Lines 384-420: complete schema detailing all of the intermediate steps (and corresponding results) in the data processing workflows for each of these reconstructions should be included.

The data processing schemes for subtomogram averaging along with intermediate steps are now illustrated in Supplementary Fig. 5. The intermediate results were previously provided as representative slices of tomograms in Extended Data Fig. 9b. We have now provided the 3-D volumes via an online repository (<https://doi.org/10.6084/m9.figshare.14527911.v2>). This part of the Methods section was also modified to enhance its clarity.

16. Lines 396-407: The separate initial reconstructions of each group (no drug, Jasplakinolide, and A23187) should be included.

We had included these subtomogram averages of individual groups (no drug, Jasplakinolide-treated and A23187-treated) in Extended Data Fig. 9b. We point to these averages now more explicitly in the Methods section and Supplementary Fig. 5. We now also provide 3-D volumes of initial RSA subtomogram averages (not utilizing the 8-fold symmetry) for each of these groups and all groups combined via an online repository (<https://doi.org/10.6084/m9.figshare.14527911.v2>).

17. In all figures, for each of the denoised tomogram sections shown, the original aligned tomograms should be provided.

We should have made this clearer – there were no denoising procedures used. All 2-D sections displayed were derived from the original tomograms. We now add:

Line 97 – *“we note that the contrast in our tomograms was sufficient to render any computational denoising unnecessary”*

18. Supplementary Videos 1 & 2: In the 360 rotating view (00:23 – 00:30 in both videos), the reconstructions look strangely flattened – almost rectangular. Do the apical ends of *C. parvum* sporozoites and *T. gondii* tachyzoites normally form a flattened tip, like the sharp end of a spade, or do they come to a rounded point, like the end of a pencil? If the latter, it is possible that there has been some compression of the tips of the parasites during vitrification, perhaps due to the surface tension from the buffer after blotting. This could affect many of the distance measurements made in the manuscript, and should be acknowledged in the main text.

The tips of the parasites are tapered and rounded. There is some flattening of these parasites on the grid that is likely due to blotting induced compression of the cytoplasm and pellicle. However, this did not appear to affect the shape of smaller organelles and structures contained within; they remained cylindrical. Flattening could have added noise to the relative positions of these features with respect to each other. We now state this explicitly and discuss which aspects of our interpretation might be impacted by this.

Line 355 – *“We note that parasites flattened on the grid, likely due to blotting. However, this flattening did not reflect on the shape of individual rhoptries, AV, MVs, or the RSAs. Flattening could have added noise to the relative positions of these features but their organizational patterns were evident despite the presence of such potential noise.”*

19. Extended Data Figure 3: Please provide the tomogram and reconstruction, just the single image is not sufficient – please also show the full tomographic section(s) from which these cropped images are drawn, and indicate on the full tomographic section(s) where the cropped images are drawn from. Given the relatively small sample size, caution needs to be taken in using these distance measurements for mechanistic conclusions (line 128), especially given the previously stated concerns that there may be some compression artifacts from the freezing process.

We are unsure if the reviewer is pointing to Extended Data Fig. 2 (line 128 of Supplementary information) or Extended Data Fig. 3 (line 128 of the main text). Nonetheless, we now provide full tomographic sections from which the cropped images were drawn and provide the positional information for the cropped images for both figures as requested. 3-D volumes for all these images have been provided via an online repository (<https://doi.org/10.6084/m9.figshare.14527794.v2> and <https://doi.org/10.6084/m9.figshare.14534109.v1>). We considerably expanded the dataset for measurements in Extended Data Fig. 2 as discussed earlier and validated our findings (Extended Data Fig. 2e-g). We also confirmed from raw tomograms that the rhoptry internal structures remained visibly unaffected by compression of the cell (see above). Moreover, such artifacts (if any) are expected to uniformly affect the various zones of rhoptry in an unbiased fashion, maintaining the veracity of our conclusions. Compression-induced noise in distance measurements of Extended Data Fig. 3, if any, did not result in the loss of organizational patterns.

20. Extended Data Figure 4: “MV’s were sometimes associated with free rhoptries, illustrating their compatibility for rhoptry docking...” How often is “sometimes?”

In a total of 52 tomograms analyzed, we found two examples for vesicles other than the AV showing rhoptry docking. Line 204 – “Furthermore, we observed cases of MVs associating with free rhoptries via the latter’s tip densities (2 out of 52 cells; Extended Data Fig. 4I)”.

21. Is the RSA present in 100% of the parasites imaged?

Yes. We now state this explicitly.

Line 237 – “All tomograms of *C. parvum* and *T. gondii* revealed an apical rosette along with other proteinaceous densities that link the AV to the plasma membrane ($n > 150$ and > 100 , respectively, Fig. 2b, g); we refer to this entire feature as the rhoptry secretory apparatus (RSA).”

22. Are there enough particles of the RSA to see if there are two species – open and closed? Could a 3D classification be done to determine whether multiple species/conformations are present?

3-D classification did not yield multiple conformations of the RSA. This may be due to insufficient numbers of subtomograms or suggest a uniform conformation that is likely closed. Biologically, this may not be surprising as rhoptry secretion is tightly controlled and linked to interaction with the host cell. Thus far there are no reports of small molecule or antibody triggering artificial rhoptry content release.

23. Given that these parasites are not in contact with a host cell, why would the RSA plug be open in some instances?

We think that the difference between the plugs in *C. parvum* and *T. gondii* RSAs (previously termed “closed” and “open”, respectively) could either be two different functional states or simply represent structural difference between the two species and not necessarily reflect the functional state. We have also renamed the plug as “radiating spokes” to remove any functional implications as part of addressing this concern.

Line 267 – “It is currently unclear whether this difference represents functional states of a passage opening and closing or simply a structural difference between the two species.”

24. Figure 3 is less informative than the Supplemental movies.

We thank the reviewer for the enthusiasm for the movies. Indeed, the 3D structure is appreciated best in a dynamic format. However, we also needed to provide reference in a static 2D figure for those readers with sole access to a printout or pdf file. In this context Figure 3 serves to describe the RSA structures by pointing to relevant features highlighted in the corresponding figure panels and text. We anticipate Figure 3 and the Supplementary Videos to be complementary.

25. Figure 4: Parts of this model are not sufficiently substantiated with data. It needs to be more clearly stated that much of this data comes from a small number of tomograms, which do not justify universal claims.

As suggested by the reviewer we have expanded the number of analyzed tomograms significantly and substantiated the features depicted in Figure 4. We now state the number of observations explicitly throughout the manuscript. Areas of speculation in the models are clearly marked with a question mark.

26. The question of how proteins are organized inside the rhoptries is mentioned in the introduction/abstract/etc, but is not addressed by the data presented. Only a speculation on proteins possibly binding the helical filaments in the neck is given, but there is no experiment presented aimed at investigating this hypothesis.

We have removed mention of rhoptry protein segregation from the Introduction to not mislead the reader's expectations. The discussion is now more carefully worded to match the conclusion to the evidence.

Line 153 – *“If and how these segregated luminal features are related to subcompartmentalization of rhoptry proteins and their possible role in staged rhoptry secretion and function remains to be established.”*

27. In many of the points presented above, raw tomograms need to be provided. Furthermore, larger sample sizes would need to be provided for any claims of a general mechanism common across the species.

We expanded sample sizes and now provide raw tomograms throughout the manuscript.

Reviewer #3 (Remarks to the Author):

Mageswaran et al. report on cryo-electron tomography studies of the rhoptry secretion complex of the apicomplexan parasites *Cryptosporidium parvum* and *Toxoplasma gondii*. This study delivers new structural insights into this complex including filaments lining the lumen of the rhoptry, and a rhoptry secretion apparatus with components on the internal and external face of the parasites apical tip. The study provides fine structural detail on more well-known structures of the complex including the apical vesicles (AVs), intraconoidal microtubules and their potential role in trafficking additional AVs to the apical tip in *T. gondii*. There is a high level of new structural detail provided in the study that help to unravel how the apicomplexan rhoptry secretory system works. They use a conditional knock-down of nD9, a protein associated with previously reported external membrane apical rosettes, to confirm that the RSA is associated with the apical rosettes and has a role in rhoptry secretion. The manuscript is well written and a pleasure to read. The imaging, analysis and models proposed are detailed with the data used to be build the models coming from repeat measures across the study. This is a really nice study that will lead to new and exciting apicomplexan invasion biology findings in the future.

We appreciate the reviewer's enthusiasm for the work.

Minor Comments:

-Please state the organism for the images in Extended Data Fig 6 and 10.

We now include the organism names in not just Extended Data Figs. 6 and 10 but in all figures for clarity.

References

- 1 Li, X. *et al.* Symmetrical organization of proteins under docked synaptic vesicles. *FEBS Lett* **593**, 144-153, doi:10.1002/1873-3468.13316 (2019).
- 2 Manca, F. *et al.* SNARE machinery is optimized for ultrafast fusion. *Proc Natl Acad Sci U S A* **116**, 2435-2442, doi:10.1073/pnas.1820394116 (2019).
- 3 Aquilini, E. *et al.* An Alveolata secretory machinery adapted to parasite host cell invasion. *Nat Microbiol* **6**, 425-434, doi:10.1038/s41564-020-00854-z (2021).
- 4 Beisson, J., Cohen, J., Lefort-Tran, M., Pouphe, M. & Rossignol, M. Control of membrane fusion in exocytosis. Physiological studies on a Paramecium mutant blocked in the final step of the trichocyst extrusion process. *J Cell Biol* **85**, 213-227, doi:10.1083/jcb.85.2.213 (1980).
- 5 Froissard, M., Keller, A. M., Dedieu, J. C. & Cohen, J. Novel secretory vesicle proteins essential for membrane fusion display extracellular-matrix domains. *Traffic* **5**, 493-502, doi:10.1111/j.1600-0854.2004.00194.x (2004).

Reviewers' Comments:

Reviewer #2:

Remarks to the Author:

Remarks to the Authors

The authors have done a good job of addressing many of the concerns previously raised regarding the main text, but there are several remaining issues that should be addressed.

1. Lines 20-21: The authors have done an excellent job of revising discussion of the apical rosette in lines 251 and 290 to clarify that while the most straightforward model for regulated rhoptry protein secretion is through the apical rosette, this has not yet been shown and thus remains a model. The fact that this is still a model needs to be reflected in lines 20-21 of the abstract as well.
2. Lines 23-24: Without knowledge of the identities, functions, or mechanisms of the components discussed here, "ultrastructure" would be a more appropriate descriptor than "molecular machine".
3. Line 25: change to "...appear to shape and compartmentalize..."
4. Line 25-26: Based on Figs 1-2 and Extended Data Figs 3 and 6, the rhoptry tip contacts the AV, which in turn appears to be attached to the plasma membrane by the RSA. However, the phrasing "mediates docking of the rhoptries to the parasite plasma membrane", which is used several times throughout the manuscript, suggests that the rhoptries are docked/attached directly to the plasma membrane by the AV.
5. Related to the previous point, in their itemized response, the authors agree that they have not formally demonstrated that the AV coordinates rhoptry secretion. As such, the name "rhoptry secretion apparatus" is not yet supported by the evidence shown in this work. However, the authors have shown evidence for consistent docking of the rhoptries to the AV, and attachment of the AV to the plasma membrane by the apical rosette. Perhaps "apical vesicle docking apparatus" would be a more accurate descriptor, given that there is no direct interaction observed between the rhoptries and the apical rosette? Or perhaps the combination of the apical rosette and the AV together could be termed the "rhoptry docking apparatus".
6. Line 26: change to "perhaps serving as a valve", as evidence for this has not yet been shown.
7. Line 27: as stated in point #1 above, in the absence of direct evidence of secretion through this apparatus, this remains a model and the word "secretory" should be removed from this sentence.
8. Line 31: is it possible to determine the directionality of the microtubules (ie whether they are pointing toward or away from the AV) from the tomograms? Perhaps by looking at the directionality/relative orientation of the linker densities between the MVs and the IMTs? If not, then perhaps the words "pointing toward" are not the most precise descriptors to use here.
9. Line 32: change structural to ultrastructural
10. Line 33-34: remove "reveal a new eukaryotic mechanism for molecular secretion", since the mechanism has not yet been determined.
11. Line 64-65: as the observed interactions between rhoptry tips and the AV have not yet been introduced/mentioned at this point in the manuscript, the connection between the apical rosette-AV interaction and rhoptry tip docking would not be obvious to the reader. For clarity, rephrase to: "...have revealed interactions between the AV and rhoptry tips, as well as interactions between the AV and the apical rosette that appear to be important for mediating rhoptry tip docking at the parasite plasma membrane and may play a role in secretion."

12. Lines 66-71: Questions ii – iv are leading questions and should be rephrased as follows for clarity:
a. ii: Are the apparent roles of the AV and the apical rosette in mediating rhoptry tip docking conserved among apicomplans, or even in *T. gondii* tachyzoites?

b. iii-iv: If the apical rosette does indeed mediate the secretion of rhoptry contents across the membranes of the AV and the parasite plasma membrane, how does the structural organization of the apical rosette support this function and how are multiple discharge events achieved in apicomplexans like *T. gondii*, which possess several rhoptries?

13. Line 75: change “likely” to “may”

14. Line 79: “ultrastructures” is a more appropriate descriptor than machineries.

15. Lines 80-81: it is not clear how observation of a similarly organized ultrastructure would directly suggest a conserved rhoptry discharge regulatory role without additional data. For clarity, rephrase to: “Subtomogram averaging of the RSA in both organisms revealed elaborate ultrastructures consisting of similarly organized extracellular, transmembrane and intracellular components, which may play a role in regulating rhoptry discharge, as they appear to mediate fusion of the AV and the plasma membrane.”

16. Throughout the manuscript and supplementary information, distance measurements are currently stated as $X \text{ nm} \pm Y \text{ nm}$. In the cases where Y is quite large, stating the measured distance as a range would make it easier for readers to appreciate the degree of accuracy of the measurement.

17. Line 123: restate $83 \pm 11\text{nm}$ as “72-94nm”. Additionally, given that the measured rhoptry diameters varied by 22nm, more than $\pm 13\%$, it does not seem accurate to state that the rhoptry diameter was uniform in line 123. It is also unclear whether the diameter variance stated here was that observed between different rhoptry necks, or within the same rhoptry neck. It would be helpful to state both the range in diameter observed within each individual rhoptry neck measured, as well as the range in diameters between all of the measured rhoptry necks.

18. Line 169: restate $10.7 \pm 5.1\text{nm}$ as 5 – 16nm. With such a large uncertainty in the measurement (varies by more than $\pm 47\%$), including the third significant figure does not accurately reflect the degree of accuracy of the measurement.

19. Line 216: restate $29 \pm 17 \text{ nm}$ as 12-46nm

20. Line 218: the range given is 35 – 150nm, but $79 \pm 38 \text{ nm}$ would give a range from 41-117nm – which range is correct?

21. Lines 226-227: restate $12.1 \pm 2.6\text{nm}$ as 9.5 – 14.7nm, which, again, is not very uniform (varies by more than $\pm 21\%$).

22. Line 416: in the itemized response, the authors state the helix has a pitch of $\sim 11\text{nm}$, while in the manuscript 11nm is used – please include the degree of accuracy on this measurement ($11\text{nm} \pm ?$). If the error is greater than 10%, please state it as a range rather than $11\text{nm} \pm X \text{ nm}$, and explain how it is possible to determine that the turns are regularly spaced and thus helical, given the uncertainty in the measurement of the spacing between turns.

REVIEWERS' COMMENTS

Reviewer #2 (Remarks to the Author):

Remarks to the Authors

The authors have done a good job of addressing many of the concerns previously raised regarding the main text, but there are several remaining issues that should be addressed.

We thank the reviewer for the criticism and concerns raised, and for acknowledging the work we put in to address them.

1. Lines 20-21: The authors have done an excellent job of revising discussion of the apical rosette in lines 251 and 290 to clarify that while the most straightforward model for regulated rhoptry protein secretion is through the apical rosette, this has not yet been shown and thus remains a model. The fact that this is still a model needs to be reflected in lines 20-21 of the abstract as well.

We refer to the ultrastructural architecture of the rhoptry secretion system as a whole and do not imply any details of the secretion model (i.e. through the apical rosette) – (line 20) *“Here, using cryo-electron tomography and subtomogram averaging, we report the conserved architecture of the rhoptry secretion system in the invasive stages of two evolutionarily distant apicomplexans.”*

We have removed the potential role of rhoptry secretory apparatus (RSA) and apical rosette in membrane fusion

2. Lines 23-24: Without knowledge of the identities, functions, or mechanisms of the components discussed here, “ultrastructure” would be a more appropriate descriptor than “molecular machine”.

We have removed the descriptor “molecular machine” from the abstract.

3. Line 25: change to “...appear to shape and compartmentalize...”

The suggested change has been made.

4. Line 25-26: Based on Figs 1-2 and Extended Data Figs 3 and 6, the rhoptry tip contacts the AV, which in turn appears to be attached to the plasma membrane by the RSA. However, the phrasing “mediates docking of the rhoptries to the parasite plasma membrane”, which is used several times throughout the manuscript, suggests that the rhoptries are docked/attached directly to the plasma membrane by the AV.

We have changed this section to the following – (line 24) *“an apical vesicle (AV), which facilitates docking of the rhoptry tip at the parasite’s apical region with the*

help of an elaborate ultrastructure named the rhoptry secretory apparatus (RSA); the RSA anchors the AV at the parasite plasma membrane.”

Other changes in the manuscript in line with this recommendation –

(line 159) *“An apical vesicle facilitates rhoptry docking.”*

(line 160) *“We observed an apical vesicle (AV) participating in rhoptry docking at the plasma membrane in all the tomograms of both T. gondii and C. parvum.”*

(line 804) *“Fig. 2: An apical vesicle (AV) and rhoptry secretory apparatus (RSA) together mediate docking of the rhoptry to plasma membrane.”*

5. Related to the previous point, in their itemized response, the authors agree that they have not formally demonstrated that the AV coordinates rhoptry secretion. As such, the name “rhoptry secretion apparatus” is not yet supported by the evidence shown in this work. However, the authors have shown evidence for consistent docking of the rhoptries to the AV, and attachment of the AV to the plasma membrane by the apical rosette. Perhaps “apical vesicle docking apparatus” would be a more accurate descriptor, given that there is no direct interaction observed between the rhoptries and the apical rosette? Or perhaps the combination of the apical rosette and the AV together could be termed the “rhoptry docking apparatus”.

We refer to the elaborate proteinaceous material that mediates docking of the AV to the plasma membrane as the rhoptry secretory apparatus (RSA). We do not include the AV into its components. We would like to kindly reiterate that the RSA is proven to be functionally important for secretion: the Nd9 depletion mutant is deficient in rhoptry secretion (Aquilini et al., 2021) and shows an improper RSA along with defects in AV-anchoring (this study). We therefore believe that the term “rhoptry secretory apparatus” is warranted.

6. Line 26: change to “perhaps serving as a valve”, as evidence for this has not yet been shown.

We have moved this description to the discussion – (line 270) *“It is currently unclear whether this difference represents functional states of a passage opening and closing (perhaps serving as a valve).”*

7. Line 27: as stated in point #1 above, in the absence of direct evidence of secretion through this apparatus, this remains a model and the word “secretory” should be removed from this sentence.

docking apparatus?

We used the descriptor “secretory” in its functional context i.e. we know that the RSA is important for rhoptry’s secretion (see response to point 5) although the

details of this mechanism remain to be further investigated. We strongly believe that the descriptor “secretory” is valid.

8. Line 31: is it possible to determine the directionality of the microtubules (ie whether they are pointing toward or away from the AV) from the tomograms? Perhaps by looking at the directionality/relative orientation of the linker densities between the MVs and the IMTs? If not, then perhaps the words “pointing toward” are not the most precise descriptors to use here.

We used the description “pointing towards” in a geometric sense i.e. to explain the positioning of AV along the axis of the microtubule. However, since the reviewer sees possibilities for misinterpreting it as a reference to microtubule polarity, we have removed it.

Our model proposing MV-trafficking towards the apical end would neither be proven nor disproven by a particular directionality for the microtubules since trafficking towards plus or minus end of a microtubule is possible using different classes of motor proteins. Moreover, it is not straightforward to test this model using the directionality of the linkers either since their identity is unknown. Rigorous testing of this model would require a lot more work including functional studies and molecular characterization of the components involved, many of which require new tools to be developed. We therefore believe that the additional analyses recommended by the reviewer are unnecessary for the current manuscript but will definitely be a part of a more elaborate future study investigating this particular aspect of rhoptry secretion.

9. Line 32: change structural to ultrastructural.

The recommended change has been made in new line 31.

10. Line 33-34: remove “reveal a new eukaryotic mechanism for molecular secretion”, since the mechanism has not yet been determined.

We have removed this sentence as per reviewer recommendation.

11. Line 64-65: as the observed interactions between rhoptry tips and the AV have not yet been introduced/mentioned at this point in the manuscript, the connection between the apical rosette-AV interaction and rhoptry tip docking would not be obvious to the reader. For clarity, rephrase to: “...have revealed interactions between the AV and rhoptry tips, as well as interactions between the AV and the apical rosette that appear to be important for mediating rhoptry tip docking at the parasite plasma membrane and may play a role in secretion.”

We have rephrased this to – (line 62) “*have revealed interactions between the AV and the rhoptry tips, as well as interactions between the AV and the apical rosette, which together mediate rhoptry tip docking at the parasite plasma membrane and may play a role in secretion.*”

12. Lines 66-71: Questions ii – iv are leading questions and should be rephrased as follows for clarity:

a. ii: Are the apparent roles of the AV and the apical rosette in mediating rhoptry tip docking conserved among apicomplans, or even in *T. gondii* tachyzoites?

b. iii-iv: If the apical rosette does indeed mediate the secretion of rhoptry contents across the membranes of the AV and the parasite plasma membrane, how does the structural organization of the apical rosette support this function and how are multiple discharge events achieved in apicomplexans like *T. gondii*, which possess several rhoptries?

The recommended changes have been made

13. Line 75: change “likely” to “may”

The recommended change has been made

14. Line 79: “ultrastructures” is a more appropriate descriptor than machineries.

The recommended change has been made

15. Lines 80-81: it is not clear how observation of a similarly organized ultrastructure would directly suggest a conserved rhoptry discharge regulatory role without additional data. For clarity, rephrase to: “Subtomogram averaging of the RSA in both organisms revealed elaborate ultrastructures consisting of similarly organized extracellular, transmembrane and intracellular components, which may play a role in regulating rhoptry discharge, as they appear to mediate fusion of the AV and the plasma membrane.”

We have made the following change – (line 78) *“Subtomogram averaging of the RSA in both organisms revealed elaborate ultrastructures consisting of similarly organized extracellular, transmembrane and intracellular components; these components could concertedly operate to regulate apicomplexan rhoptry discharge in an evolutionarily conserved fashion, as they appear to mediate fusion of the AV and the plasma membrane.”*

16. Throughout the manuscript and supplementary information, distance measurements are currently stated as $X \text{ nm} \pm Y \text{ nm}$. In the cases where Y is quite large, stating the measured distance as a range would make it easier for readers to appreciate the degree of accuracy of the measurement.

Taking the reviewer’s advice, we have now provided a range that represents the 95% confidence interval (C.I.) in addition to $X \text{ nm} \pm Y \text{ nm}$ ($X = \text{mean}$, $Y = \text{std}$) for cases where the reviewer has pointed out large Y values. We have chosen the 95% C.I. instead of the option provided by the reviewer i.e. (mean minus std)-to-(mean plus std) since the former is more statistically relevant and reflects the accuracy/confidence in our measurements, which is the main reason for this reviewer request. Also, the actual ranges for all our measurements are directly evident from the Supplementary Figure graphs. For consistency, we have also retained $X \text{ nm} \pm Y \text{ nm}$ values throughout the manuscript.

17. Line 123: restate $83 \pm 11\text{nm}$ as “72-94nm”. Additionally, given that the measured rhoptry diameters varied by 22nm, more than $\pm 13\%$, it does not seem accurate to state that the rhoptry diameter was uniform in line 123. It is also unclear whether the diameter variance stated here was that observed between different rhoptry necks, or within the same rhoptry neck. It would be helpful to state both the range in diameter observed within each individual rhoptry neck measured, as well as the range in diameters between all of the measured rhoptry necks.

We have now provided the range for rhoptry diameters within individual rhoptries and within several rhoptries taken together (Supplementary Table 1, Supplementary Fig. 1h).

Line 123 – “*mean \pm std = $83 \pm 11\text{ nm}$ and 95% confidence interval or C.I. = $82 - 83\text{ nm}$ ”*

Line 126 – “*This uniformity in rhoptry diameter was even more evident from individual rhoptries (Supplementary Table 1).*”

We have retained the description “uniform” since the newly calculated 95% C.I. for all rhoptry neck measurements (82-83 nm) supports our conclusion. It is worth noting that the large standard deviation (11 nm) stems mainly from a couple of rhoptries (rhoptries 10 and 12; Supplementary Table 1). Moreover, individual rhoptry necks show a very uniform diameter.

18. Line 169: restate $10.7 \pm 5.1\text{nm}$ as 5 – 16nm. With such a large uncertainty in the measurement (varies by more than $\pm 47\%$), including the third significant figure does not accurately reflect the degree of accuracy of the measurement. We have now provided the range (95% C.I.) to indicate accuracy/confidence in our measurements. It is worth noting that the voxel size in our tomograms (after 4 times binning) is 1 nm. We therefore believe that the variability in our measurements reflects the true variability in the data to a large extent although minor inaccuracies are possible in the range of 1 nm.

Line 170 – “*The rhoptries were associated with the AV at a distance of $10.7 \pm 5.1\text{ nm}$ (95% C.I. = $9.2 - 12.2\text{ nm}$)”*

19. Line 216: restate $29 \pm 17\text{ nm}$ as 12-46nm

We have now provided the range – (line 217) “*The most apical MV lined up close to the apical tip of the IMT ($L_1 = 29 \pm 17\text{ nm}$ with a 95% C.I. of $23 - 35\text{ nm}$; Supplementary Fig. 8f)*”

20. Line 218: the range given is 35 – 150nm, but $79 \pm 38\text{ nm}$ would give a range from 41-117nm – which range is correct?

We had originally provided a range based on a quick visual observation of Supplementary Fig. 8f. However, taking the reviewer comment into consideration, we have now provided the 95% C.I. for this distribution – (line 219) *“The posterior region of the IMT unoccupied by MVs (L_{last}) was the most variable (79 ± 38 nm with a 95% C.I. of 65 – 93 nm; considerably larger in comparison to L_1 but not exceeding the required space for 2 MVs).”*

21. Lines 226-227: restate 12.1 ± 2.6 nm as 9.5 – 14.7nm, which, again, is not very uniform (varies by more than $\pm 21\%$).

Upon calculating the 95% C.I. our range was quite narrow (11.5 – 12.7 nm) and we therefore retain the descriptor “uniform”. Some of the variability in the data could be from minor measurement errors (note that the pixel size of 1 nm is somewhat close to the standard deviation reported). Therefore the 95% C.I. in such cases tells us about the certainty in measurements (notwithstanding the actual variability in data).

Line 227 – *“Our tomograms revealed that the distances between the MVs and their associating IMTs were uniform (12.1 ± 2.6 nm with a 95% C.I. of 11.5 – 12.7 nm; Supplementary Fig. 7f-h)”*

22. Line 416: in the itemized response, the authors state the helix has a pitch of ~11nm, while in the manuscript 11nm is used – please include the degree of accuracy on this measurement ($11\text{nm} \pm ?$). If the error is greater than 10%, please state it as a range rather than $11\text{nm} \pm X$ nm, and explain how it is possible to determine that the turns are regularly spaced and thus helical, given the uncertainty in the measurement of the spacing between turns.

We have now provided a more precise pitch for the helical filaments in question as mean \pm std and 95% C.I. The pitch is indeed quite uniform close to 11 nm. The small variability can be explained by both flexibility in these filaments (rhostry neck contours are not always straight) and minor errors in measurements and do not change any of our previous conclusions.

Line 425 – *“an ~11 nm pitch (11.1 ± 0.7 nm with a 95% C.I. of 10.9 – 11.3 nm)”*

The helical geometry of this filament was evident from not just the regular spacing between the filament turns but also from the continuity of the helical filament densities in the 3-D volume (Supplementary Fig. 4 and Supplementary Movie 3).